# Heterogeneity in lung macrophage control of *Mycobacterium tuberculosis* is modulated by T cells

Rocky Lai [1], Travis Williams[1], Tasfia Rakib[1], Jinhee Lee[1] & Samuel M. Behar [1] ✉

Following *Mycobacterium tuberculosis* infection, alveolar macrophages are initially infected but ineffectively restrict bacterial replication. The distribution of *M. tuberculosis* among different cell types in the lung changes with the onset of T cell immunity when the dominant infected cellular niche shifts from alveolar to monocyte-derived macrophages (MDM). We hypothesize that changes in bacterial distribution among different cell types is driven by differences in T cell recognition of infected cells and their subsequent activation of antimicrobial effector mechanisms. We show that CD4 and CD8 T cells efficiently eliminate *M. tuberculosis* infection in alveolar macrophages, but they have less impact on suppressing infection in MDM, which may be a bacterial niche. Importantly, CD4 T cell responses enhance MDM recruitment to the lung. Thus, the outcome of infection depends on the interaction between the T cell subset and the infected cell; both contribute to the resolution and persistence of the infection.

*Mycobacterium tuberculosis* causes the chronic lung infection tuberculosis (TB). Following inhalation of *M. tuberculosis*, alveolar macrophages are initially infected, setting in motion a series of events that result in the recruitment of numerous cell types to the lung, including both innate (e.g., neutrophils and macrophages) and adaptive (e.g., B and T cells) immune cells[1–3]. These cells cooperate to form granulomas, the characteristic pathological lesion of TB. The fate of granulomas is varied. Some undergo fibrosis and/or calcification. In others, progressive bacterial replication leads to dissemination, necrosis, and cavitation. While most people develop immunity and avoid symptomatic disease, 10% develop clinical TB at some point during their lives. Impairment of cell mediated immunity greatly increases the risk of developing active TB, and genetic, environmental, and microbial factors play a role. Understanding how *M. tuberculosis* is contained is an important question as a major clinical goal is to develop vaccines that prevent susceptible people from developing disease. A better understanding of the mechanisms that lead to containment of infection and why they fail in susceptible individuals is needed.

Tremendous heterogeneity exists among cells in the myeloid lineage. Monocytes, macrophages, dendritic cells (DC), and polymorphonuclear cells (PMNs) (hereafter referred collectively to as myeloid cells) both reside in and are recruited to the lung parenchyma during *M. tuberculosis* infection. Heterogeneity is based on cell ontology but is also shaped by homeostatic signals specific for the tissue niches where the cells reside (e.g., alveolar macrophages), and by episodic inflammatory signals that lead to cell recruitment, activation, and differentiation. After inhalation of aerosols containing *M. tuberculosis*, alveolar macrophages (AM) are the first cell type in the lung infected[1]. Subsequently, AM in the alveoli traffic into the parenchyma and trigger innate inflammatory responses. Monocytes and neutrophils are recruited to the lung. Monocytes differentiate into macrophages and dendritic cells[4,5], which *M. tuberculosis* infects[1,6–9]. New experimental approaches show that pre-existing heterogeneity among human macrophages affect *M. tuberculosis* growth[10]. Importantly, macrophages differ in their intrinsic control of *M. tuberculosis* in vivo, which may be triggered by cell-specific responses[11–13]. In particular, AM have a metabolic environment that is conducive to bacillary growth by serving as a source of iron and fatty acids, while glycolytically-biased monocyte-derived macrophages (MDM) are more restrictive[11,12,14]. While AM are the initial cell type infected after aerosol *M. tuberculosis*

[1]Department of Microbiology and Physiological Systems, University of Massachusetts Medical School, Worcester, MA, USA.
✉e-mail: samuel.behar@umassmed.edu

infection[1,12], after infection is established, the number of infected CD11c+ monocyte-derived cells (hereafter, MDM) outnumbers infected AM[1,9]. While intrinsic features of infected myeloid cells in the lung may affect their propensity to sustain or restrict *M. tuberculosis* growth early after infection, we hypothesize that once adaptive immunity is initiated and recruited to the lungs, T cell immunity will modify the capacity of myeloid cells to restrict *M. tuberculosis* replication.

In animal models, *M. tuberculosis* is controlled within weeks after infection and bacterial exponential growth is replaced by a plateau phase. The timing of this transition is coincident with the development of T cell immunity. The use of MHCII+/+/MHCII-/- mixed bone-marrow chimeric mice provides direct evidence that cognate interactions between CD4 T cells and infected cells is important for containment of infection in vivo, although Mtb was not eliminated from MHCII+ lung macrophages[15]. Infected CD11c+ MDM are highly activated, express NOS2, and upregulate CD14, CD38 and ABCA1, to a greater degree than their uninfected counterparts, providing additional evidence that infected cells interact with T cells[9]. Yet, it is paradoxical that MDM should be a dominant reservoir of *M. tuberculosis* and show evidence of being activated by T cells. These data suggest that T cell control of Mtb-infected macrophages is suboptimal and raises the possibility of a cellular niche that supports continued bacillary persistence.

Not all *M. tuberculosis*-specific T cells efficiently recognize *M. tuberculosis*-infected macrophages, depending on the antigen[16,17] and the number of bacilli per macrophage[18–21]. It is unknown whether the interaction of CD4 and CD8 T cells with *M. tuberculosis*-infected cells varies depending on the type of infected cell. We hypothesize that some *M. tuberculosis* bacilli occupies a protected cellular niche. We reasoned that for cell types in which *M. tuberculosis* could be eliminated or its growth restricted by T cells, T cell depletion would lead to an increase in infected cells. In contrast, for cell types in which *M. tuberculosis* persists, even in the face of T cell pressure, T cell depletion should have minimal effect. Here we report that CD4 and CD8 T cells cooperate to restrict *M. tuberculosis* infection in several types of infected cells. *M. tuberculosis*-infection in AM is efficiently controlled by T cells, particularly by CD4 T cells. In contrast, both CD4 and CD8 T cells are required to restrict *M. tuberculosis* replication in MDM. Interestingly, T cell depletion had the least effect on MDM compared to other cell types. This hierarchy is the reverse of the intrinsic capacity of AM and MDM to control bacterial replication early after infection. Thus, depending on the cell type, the influence of T cells on *M. tuberculosis* containment varies. We propose that although T cell interact with *M. tuberculosis*-infected MDM, they have a limited ability to restrict infection in these cells. Although MDM are better than AM in their intrinsic capacity to restrict intracellular *M. tuberculosis*, once T cell immunity is initiated, AM largely inhibit *M. tuberculosis* growth while T cells contribute only modestly to the ability of MDM to limit intracellular infection.

## Results

### CD4 and CD8 T cells synergize to restrict *M. tuberculosis* growth in the lung

To determine how T cells exert pressure on different types of *M. tuberculosis*-infected macrophages, we combined low dose aerosol infection with Rv.YFP[9] and antibody-mediated depletion of T cells. The in vivo infection was allowed to progress for three weeks, during which time T cell immunity to *M. tuberculosis* is initiated and recruited to the lung[2,3]. Then, groups of mice were treated with an antibody to CD4, to CD8, or a combination of anti-CD4 and anti-CD8 for two weeks. The mice were analyzed 5 weeks post infection (wpi) (Fig. 1a). Both CD4 and CD8 T cells were eliminated from the lungs of *M. tuberculosis* infected mice, although CD8 depletion was slightly less efficient (Fig. 1b). Neither CD4 nor CD8 T cell depletion led to a statistically significant increase in lung CFU (Fig. 1c). In contrast, the combined depletion of

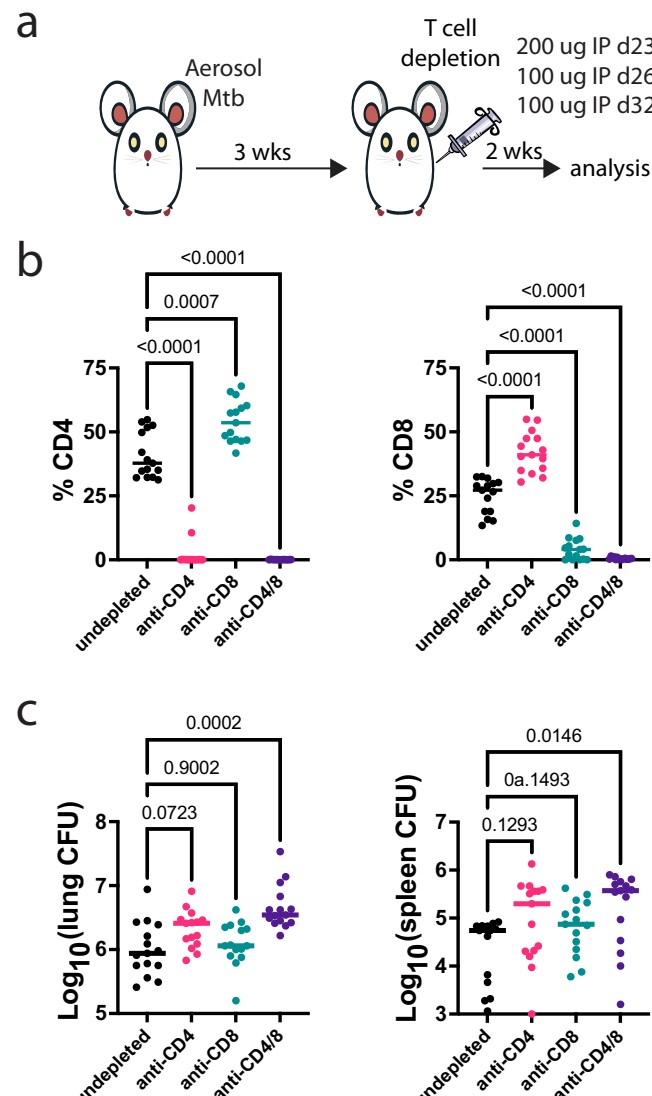

**Fig. 1 | The effect of CD4 and CD8 T cell depletion on lung and spleen CFU.** **a** Experimental scheme. Mice were infected with Rv.YFP and rested for 3 weeks, after which CD4 and CD8 depleting mAbs were given over the course of two weeks at the indicated timepoints. **b** Proportion of CD4 and CD8 T cells in the lungs of infected C57BL/6 mice at 5 weeks post infection following treatment with either anti-CD4 and/or anti-CD8 depleting mAbs. **c** Lung and spleen CFU at 5 weeks post infection following treatment either anti-CD4 and/or anti-CD8 depleting mAbs. Each point represents an individual mouse. The data are from n = 15 mice, pooled from three independent experiments. Each condition was compared to the undepleted group using Welch's ANOVA test followed by the Dunnett multiple comparison test. Bar, median. *P*-values are indicated. Source data are provided as a Source Data file.

CD4 and CD8 T cells led to an increase in lung CFU (Fig. 1c). These data show that early after infection, both CD4 and CD8 T cells make a synergistic contribution to controlling *M. tuberculosis* replication in vivo.

### Monocyte-derived macrophages are the dominant cellular niche for *M. tuberculosis*

Advances in multiparametric flow cytometry have improved our ability to characterize myeloid cells in the lung. We adapted our myeloid flow panel for spectral flow cytometry and measured the distribution of *M. tuberculosis* among different myeloid cell types in the lung, 3- and 5-weeks post-infection. Some important technical features of the panel

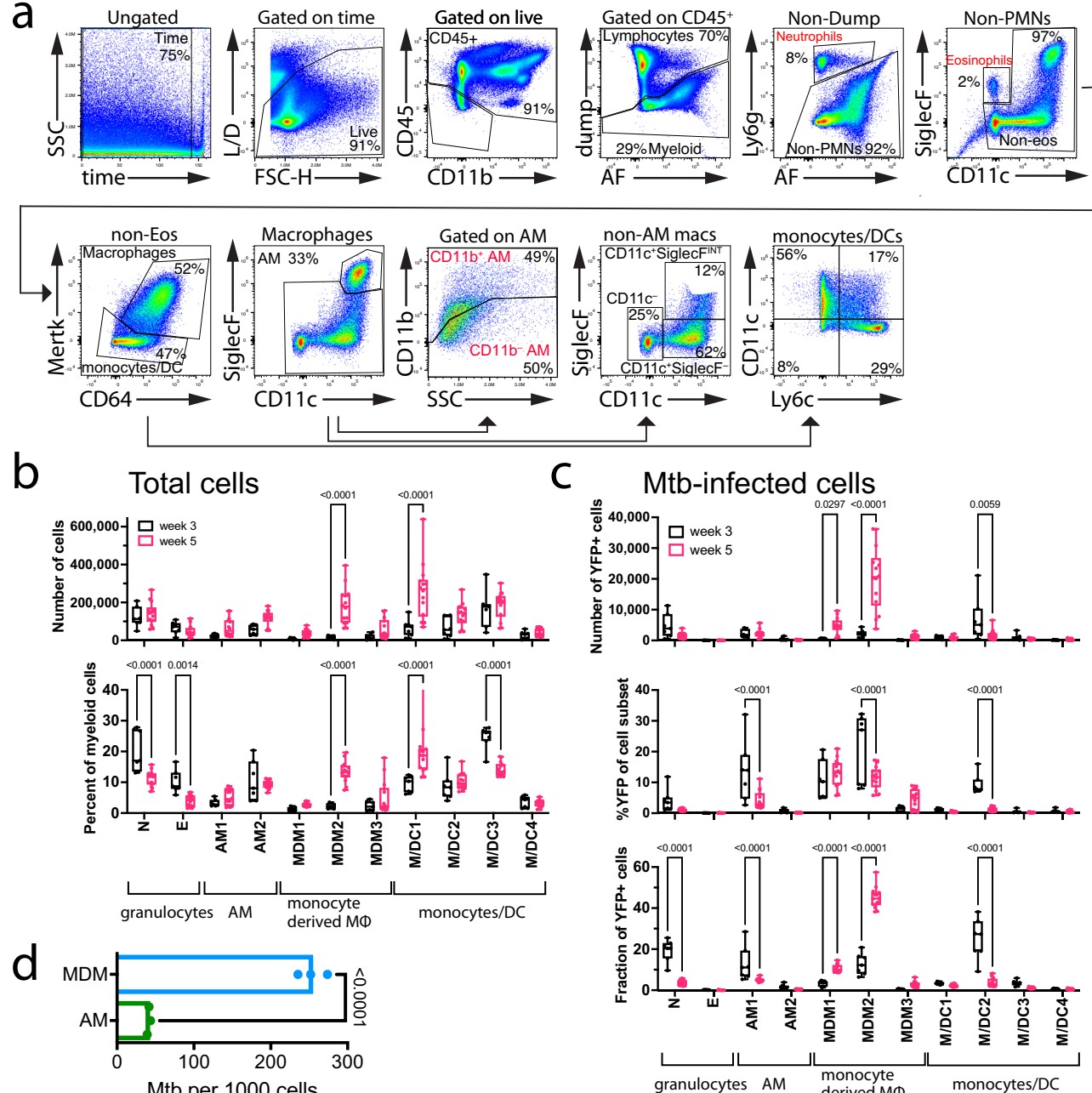

**Fig. 2 | Distribution of *M. tuberculosis*-infected cells between three and five weeks after infection. a** Gating strategy for identifying lung myeloid cell populations. In brief, a viability dye was used to exclude dead cells; CD45 and CD11b were used to exclude non-hematopoietic cells; CD3, CD19 and NK1.1 (vs. autofluorescence, AF) were used to exclude lymphocytes from other hematopoietic cells. Ly6G and AF allowed identification of neutrophils; similarly, CD11c and SiglecF identified eosinophils. Mertk and CD64 were used to separate macrophages from monocyte/DC subsets. SiglecF and CD11c were used to separate AM from MDM, and then used to subset the MDM populations. CD11b was used to divide the AM population. CD11c and Ly6C were used to subset the monocyte/DC populations. **b** Quantification of different myeloid cell populations in the lung at week 3 and week 5 post infection, expressed in terms of total cell numbers (top) or as a percentage of total myeloid cells (bottom). **c** Quantification of the YFP signal in various

myeloid cell types in the lung at week 3 and week 5 post infection, expressed in terms of total cell numbers (top), %YFP of each population (middle) or the fraction of each population within the total population of YFP⁺ cells (bottom). **d** Flow sorted AM and MDM2 were plated to determine intracellular CFU within each purified population. **b, c** Each point represents an individual mouse pooled from two independent experiments ($n = 5$) for week 3, and three independent experiments ($n = 15$) for week 5. A two-way ANOVA was performed using Bonferonni's multiple comparison test. Box plots indicate median (middle line), 25th, 75th percentile (box) and minimum and maximum (whiskers). *P*-values are indicated. **d** Results are representative of two experiments using cells purified from lung cells pooled from five mice, tested in triplicate, and analyzed using a t-test. Bars are mean ± SD. P (two-sided), $p < 0.0001$. Source data are provided as a Source Data file.

and its analysis are described in the *Methods*. Myeloid cells were defined as live CD45⁺ cells after lymphoid cells were excluded based on the CD19, Thy1.2, and NK1.1 lineage markers. Neutrophils and eosinophils were identified by their expression of Ly6G and SiglecF,

respectively (Fig. 2a). CD64 and Mertk distinguished macrophages from non-macrophages (i.e., monocytes and DC).

Alveolar macrophages (AM) were discriminated from other lung macrophages by their high levels of SiglecF and CD11c. CD11b

expression divided AM into two subsets. Non-AM macrophages have been called recruited macrophages (RM), interstitial macrophages (IM) and CD11c[Hi] monocyte-derived cells (MDC)[5,9,11,12]. We previously referred to these cells as CD11c[Hi]; however, in recognition of heterogeneity in their CD11c expression, we have dropped the CD11c moniker. As these monocyte-derived cells are distinct from resident macrophages (e.g., AM), we refer to them as monocyte-derived macrophages (MDM). MDM were divided into three subsets based on their SiglecF and CD11c expression. The SiglecF[int]CD11c[+] (MDM1) were the most variable between experiments and could be immature AM[22–24]. SiglecF[−] (MDM2) were the most abundant of the three and were most like what we previously referred to as CD11c[Hi] MDC (Fig. 2a)[9]. In additions, SiglecF[−]CD11c[−] (MDM3) may be nerve associated macrophages that have been recently described in the lung[25]. Finally, we subdivided monocytes and DC (M/DC) based on CD11c, Ly6C, CD26, CD11b and MHCII expression (M/DC1-4). (Supplementary Fig. 1). The most abundant of these were M/DC1 (Ly6C[−]CD11c[VAR]CD26[+]CD11b[var]MHCII[hi]) and M/DC3 (Ly6C[+]CD11c[−]CD26[-]CD11b[+]MHCII[low]). The former was likely a mixed DC population, and the latter were probably classical monocytes. M/DC2 (Ly6C[+]CD11c[+]CD26[+]CD11b[+]MHCII[hi]) are likely a monocyte-derived DC population based on Ly6C expression.

Between three and five weeks after infection, the total number of macrophages and monocyte/DCs in the lung significantly increased (Supplementary Fig. 2a). During this interval, the number of *M. tuberculosis*-infected macrophages increased 5.4-fold, while the number of infected eosinophils, neutrophils, and monocyte/DCs remained the same. This led to macrophages becoming the dominant infected cell type (Supplementary Fig. 2b, c). With the greater granularity afforded by spectral flow cytometry, we defined what was driving these changes in 11 predefined cell subsets (Fig. 2a). Between three and five weeks after infection, SiglecF[−]CD11c[+] non-AM macrophages (i.e., MDM2) underwent a 14-fold increase in cell number such that they accounted for ~14% of the lung myeloid cells (Fig. 2b). M/DC1 cells (Ly6C[−]CD11c[+]), also increased in number (Fig. 2b). Thus, the dominant myeloid cell types in the lung changed from predominantly neutrophils and monocytes (M/DC3) to macrophages (MDM2) and DC (M/DC1). This was accompanied by a dramatic shift in the type of cells infected by *M. tuberculosis* (Fig. 2c). The number of *M. tuberculosis*-infected non-alveolar macrophages (MDM1, −2, −3) all increased by more than 10-fold. In absolute numbers, MDM2 increased the most and came to account for 45% of infected cells (Fig. 2c). To confirm that these results were not due to differences in YFP expression between the different macrophage populations, we determined the intracellular CFU within AM and MDM from the lungs of infected B6 mice. Consistent with our flow cytometric analysis (Fig. 2a, c), MDM had 10-fold more *M. tuberculosis* CFU than AM (Fig. 2d). The significant increase in *M. tuberculosis*-infected macrophage number occurred despite the onset of T cell immunity during this phase of infection[3,26]. We hypothesized that MDM could represent a cellular niche against which T cell immunity inefficiently controlled *M. tuberculosis* replication.

### The effectiveness of T cell immunity depends on the type of infected cell

We predicted that a cellular niche that was shielded from T cell immunity would be largely unaffected by T cell depletion, compared to cell types that require T cell signals to restrict intracellular *M. tuberculosis* replication. As described above, three weeks after infection, once effector T cells were recruited to the lung, groups of mice were treated with mAb to CD4, CD8 or both CD4 and CD8 for two weeks. Then, the frequency of *M. tuberculosis*-infected cells was determined for 11 myeloid cell types by flow cytometry (Figs. 1a, 2a).

T cell depletion led to a statistically significant increase in the percentage of infected neutrophils, AM1, MDM1, and MDM2 (Fig. 3a, b). Only combined CD4 and CD8 depletion led to an increased frequency of *M. tuberculosis*-infected neutrophils, suggesting that CD4

and CD8 T cells are redundant in their ability to control infection in these cells (Fig. 3b). In contrast, CD4 depletion alone led to a significantly increased frequency of infected AM1, MDM1, and MDM2 (Fig. 3a, b). Although CD8 depletion alone had no effect, CD4 and CD8 double depletion significantly increased the frequency of infected AM1, MDM1, and MDM2 compared to CD4 depletion alone (Fig. 3a, b). T cell depletion had minimal impact on the viability of the myeloid cells in the lung (Supplementary Fig. 3a). Importantly, these data indicate CD4 and CD8 T cells act synergistically to control *M. tuberculosis* infection in macrophages in vivo. Only depletion of CD4 and CD8 T cells led to an increase in the number of bacilli per cell in AM1, AM2, and MDM1, based on the median fluorescence intensity of Rv.YFP (Fig. 3c). Interestingly, the Rv.YFP signal of MDM2, the most abundant infected cell type, was unaffected by T cell depletion (Fig. 3c). However, CD4 depletion led to a decrease in the number of MDM (Supplementary Fig. 3b, c). We also confirmed the lack of CD4 or CD8 expression by lung myeloid cells (Supplementary Fig. 4), demonstrating that our depletion protocol did not result in the direct depletion of myeloid cells.

We next analyzed how T cell pressure affected the frequency of infected cells among all myeloid cells, since the abundance of each cell type varies. In control (undepleted) mice, MDM2 made the largest contribution (Fig. 3d). CD4 T cell significantly augmented the fraction of infected neutrophils and AM1 among myeloid cells (Fig. 3b, d). In contrast, CD4 T cell depletion did not alter the contribution of infected MDM2. While CD8 depletion alone had no effect, combined anti-CD4 and anti-CD8 mAb treatment significantly increased the frequency of infected neutrophils and M/DC2, compared to anti-CD4 or CD8 mAb treatment alone. Combined CD4 and CD8 depletion also increased the fraction of infected MDM1 and MDM2. However, the contribution of MDM2 to *M. tuberculosis*-infected myeloid cells after CD4 + CD8 depletion increased 1.4-fold (compared to undepleted mice). This change was small compared to the increased proportion of infected neutrophils or AM1 (9.5- and 5.8-fold, respectively). Thus, even though MDM are highly activated, their high rate of infection and dominant niche for *M. tuberculosis* raise the possibility that they are inefficiently recognized by T cells[16,18,27]. Conversely, the emergence of other cell populations as an important niche for *M. tuberculosis* after T cell depletion shows that T cell immunity effectively controls *M. tuberculosis* infection in other myeloid cell types.

### CD4 T cells maintain high levels of NOS2 expression by macrophages

Nitric oxide synthase 2 (NOS2) is induced by IFNγ in macrophages and its expression is essential for survival of mice after *M. tuberculosis* infection. NOS2 converts L-arginine into nitric oxide (NO), which is toxic to *M. tuberculosis*; however, its role in vivo is more complicated as is its relevance to human TB[28]. NOS2 is expressed by *M. tuberculosis*-infected macrophages in the lung lesions of B6 mice, both by AM and MDM[9,12,29]. We measured NOS2 expression by myeloid cells in the lung following *M. tuberculosis* infection and T cell depletion as described above. At baseline (i.e., undepleted), 25–42% of MDM1 and MDM2 expressed NOS2 (Fig. 4a). Other cell types such as AM1, MDM3, and M/DC2 also produced significant amounts of NOS2. The NOS2 expression by these different cell types correlated with their degree of infection (Fig. 4b, $r = 0.98$). In general, more cells in each population expressed NOS2 than were infected. This led us to determine how many *M. tuberculosis*-infected cells expressed NOS2 (Fig. 4c). There were too few eosinophils and AM2 to generate reliable data. However, for the other subsets, there was a hierarchy of NOS2 producing cells. Nearly 100% of the *M. tuberculosis*-infected MDM subsets expressed NOS2. In contrast, ~70% of AM1, 50–60% of M/DC except for M/DC3 produced NOS2. We did not detect intracellular NOS2 in neutrophils (Fig. 4c).

We next looked at the effect of T cells on the expression of NOS2 in *M. tuberculosis*-infected cells by segregating the frequency of NOS2[+]

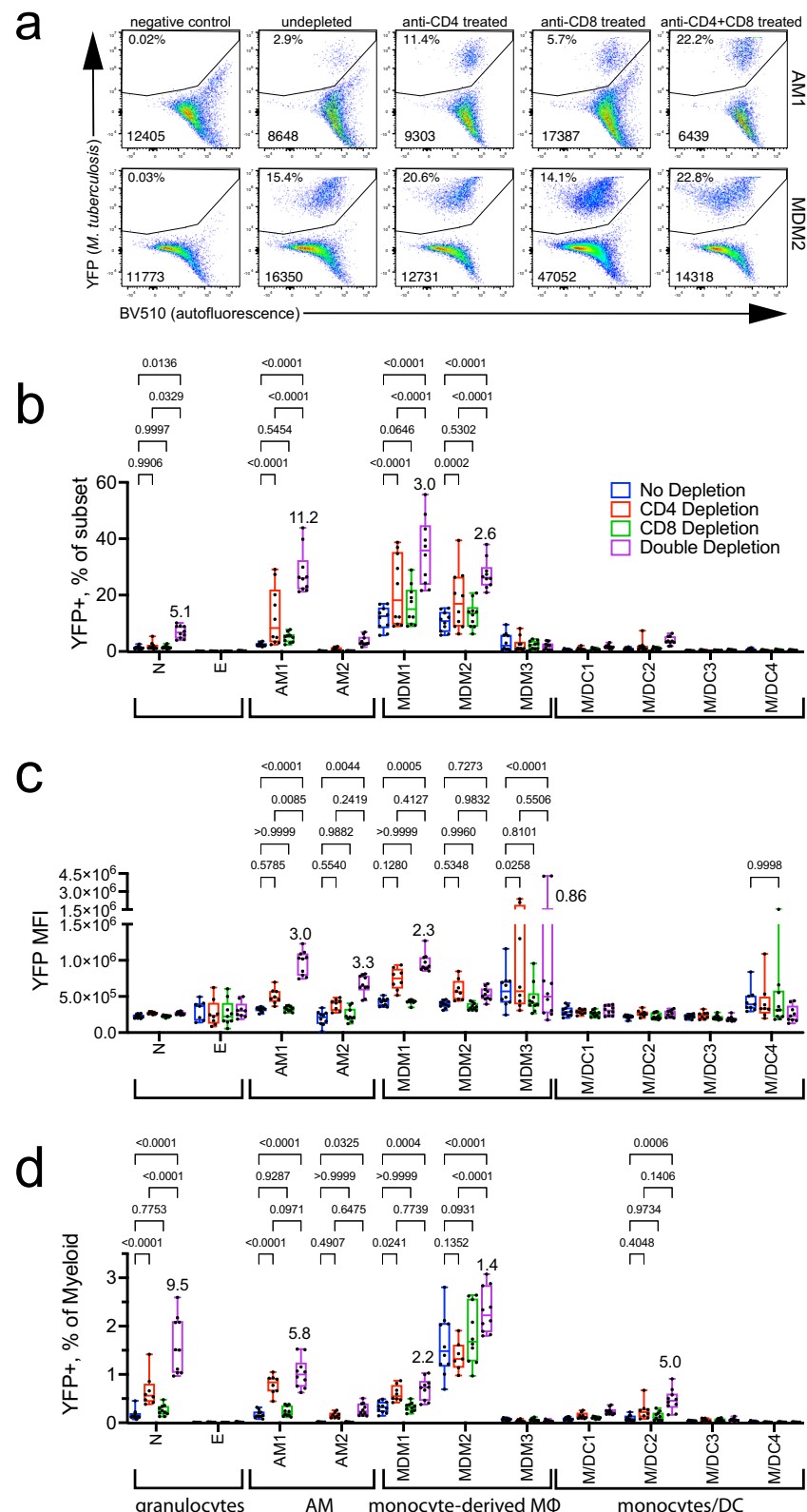

**Fig. 3 | Different types of lung myeloid cells react differently to the loss of CD4 and CD8 T cell pressure.** Quantification of YFP signal within various myeloid cell populations in the lung at 5 weeks post infection following CD4 and/or CD8 depletion. **a** Representative flow plots of YFP signal in AM1 and MDM2, plotted against autofluorescence. Top number represents the frequency of YFP⁺ events, bottom number are the total number of cells. **b**–**d** Graphical representation of YFP signal in different myeloid subsets, expressed as (**b**) %YFP+ cells within each subset; **c** MFI of YFP within each subset; and (**d**) YFP⁺subset⁺ cells as a percentage of the

total myeloid cells. Each point represents an individual mouse from two independent experiments (*n* = 10). A two-way ANOVA was performed using Tukey's multiple comparison test. Box plots indicate median (middle line), 25th, 75th percentile (box) and minimum and maximum (whiskers). Some statistical comparisons have been omitted for clarity. Numbers in (**b**–**d**) are the fold change of the 'double depletion' compared to the 'no depletion' condition. Source data are provided as a Source Data file.

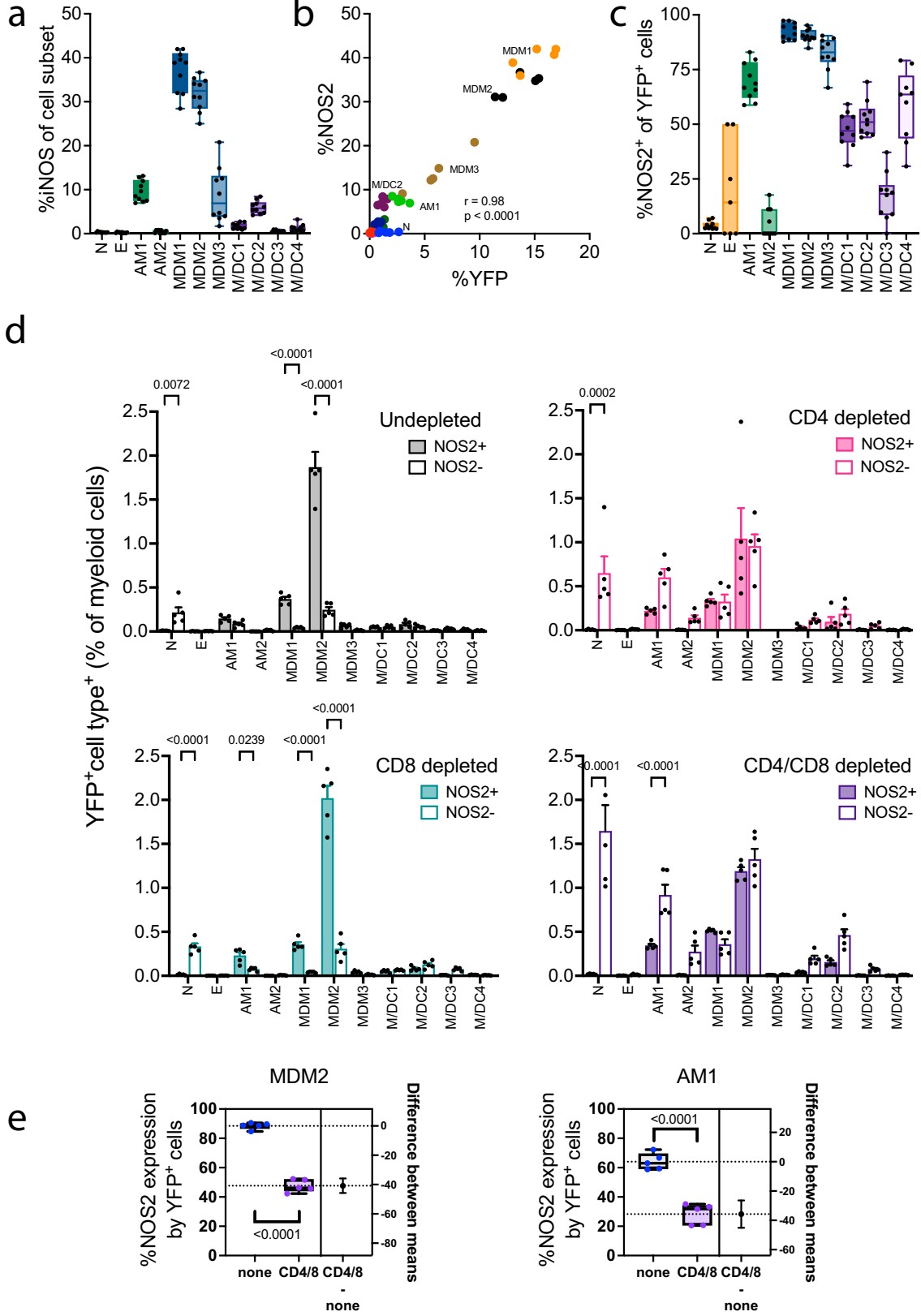

**Fig. 4 | T cell depletion affects NOS2 expression. a** The percentage of different myeloid cell types in the lung that express NOS2 at 5 wpi. **b** Pearson correlation between %NOS2 expression and %YFP signal in various myeloid cell populations. **c** The percentage of infected (YFP+) cells that express NOS2. **d** The effect of CD4 and/or CD8 T cell depletion on the distribution of YFP+ cells among NOS2+ and NOS2− cells, as a percentage of total myeloid cells in the lung at 5 wpi. A two-way ANOVA was performed using Šídák's multiple comparisons test. Bars are mean ± SEM. *P*-values are indicated. **e** The effect of CD4 and CD8 T cell depletion on the frequency of NOS2 expression by YFP+ AM1 and MDM2 in the lung at 5 wpi. T-test. *P*-values (two-tailed) are indicated. The data (*n* = 5) is one of two independent experiments. **a**, **c**, **e** Box plots indicate median (middle line), 25th, 75th percentile (box) and minimum and maximum (whiskers). Source data are provided as a Source Data file.

vs. NOS2⁻ infected cells. After CD4 depletion, slightly more than half of the *M. tuberculosis*-infected cells failed to express NOS2. CD4 depletion led to an increase in *M. tuberculosis*-infected neutrophils and AM1, and most infected cells failed to express NOS2 (Fig. 4d). Consistent with our previous results, CD8 depletion did not affect the frequency of *M. tuberculosis*-infected cells nor their expression of NOS2, although it modestly increased the number of infected neutrophils. The ratio of NOS2⁺ vs. NOS2⁻ infected cells was decreased after CD4 and CD4 + CD8 depletion in all myeloid populations (Fig. 4d). Remarkably, although ~50% of the *M. tuberculosis*-infected MDM2 no longer expressed NOS2 after CD4 or CD4/CD8 depletion, the frequency of infected cells only increased incrementally (Fig. 4d). While infected MDM2 express more NOS2 than AM1 at baseline (i.e., in an intact B6 mouse), the effect of CD4 + CD8 T cell depletion leads to a similar absolute reduction in NOS2 expression by YFP⁺ AM1 and MDM2 (Fig. 4e). Although NOS2 continues to be expressed in myeloid cells even in the absence of T cells, these data indicate that CD4 T cells are crucial for maintaining high levels of NOS2 expression in infected macrophages.

## Aminoguanidine treatment does not affect control of *M. tuberculosis* infection

NOS2 is essential for resistance to *M. tuberculosis* infection in B6 mice and mice that lack the NOS2 gene succumb to infection 4-5 wpi[30]. NO has been suggested to be more important in regulating inflammation than in direct killing of *M. tuberculosis*[31,32]. We tested whether inhibition of NOS2 in vivo, through administration of aminoguanidine (AG), would exacerbate *M. tuberculosis* infection as observed with CD4 T cell depletion (Fig. 5a). To verify that AG treatment was successful, NO was measured in lung homogenate using the Griess reaction. AG appeared to have successfully inhibited the production of NO by NOS2 (Fig. 5b). We observed an increase in the number of viable bacilli in the lungs but not the spleens of mice treated with AG for two weeks (Fig. 5c). The total number of neutrophils and M/DC3 but not any of the other myeloid populations were significantly increased after treatment (Fig. 5d). The frequency of infected neutrophils was increased after AG treatment, consistent with the exacerbation of TB disease as previous described[32]. However, AG treatment did not alter the proportion of infected cells among the other myeloid subsets (Fig. 5e). Thus, it appears that this early stage of *M. tuberculosis* infection after T cell recruitment to the lung, NO has little or no role in the control of *M. tuberculosis* within macrophage subsets.

## CD4 T cells drive the recruitment of MDM to the site of infection

Our data suggests that AM depend on CD4 T cells to augment their intrinsic capacity to control *M. tuberculosis* (Fig. 3a). In contrast, MDMs are less sensitive to the loss of T cell pressure. T cells secrete numerous chemokines including CCL2, CCL3, and CCL4, which direct cellular recruitment to the site of infection. As such, we postulated that CD4 T cells recruitment of myeloid cells to the lung could augment containment of *M. tuberculosis* infection. To determine whether T cells were important in the recruitment of myeloid cells, C57BL/6 and RAG1 knockout (KO) mice, the latter being devoid of B and T cells, were infected with *M. tuberculosis*. At 3 weeks post infection, CD4 T cells from infected C57BL/6 mouse were transferred via the intravenous route to the RAG1 KO mice (Fig. 6a). As expected, transfer CD4 T cells were sufficient to protect help contain *M. tuberculosis* (Fig. 6b). *M. tuberculosis* infection of RAG1 KO mice led to an increase in many of the different myeloid cell types in the lung (Fig. 6c). Transfer of immune polyclonal CD4 T cells led to a large significant increase in the number of MDM in the lung but not in other myeloid cell populations (Fig. 6c). Immune polyclonal CD4 T cells reduced the fraction of *M. tuberculosis* infected cells (Fig. 6d, left) and decreased the *M. tuberculosis* bacillary load, based on YFP MFI (Fig. 6d, right), for most of the myeloid cell types in the lung. Given these effects of CD4 T cells, one

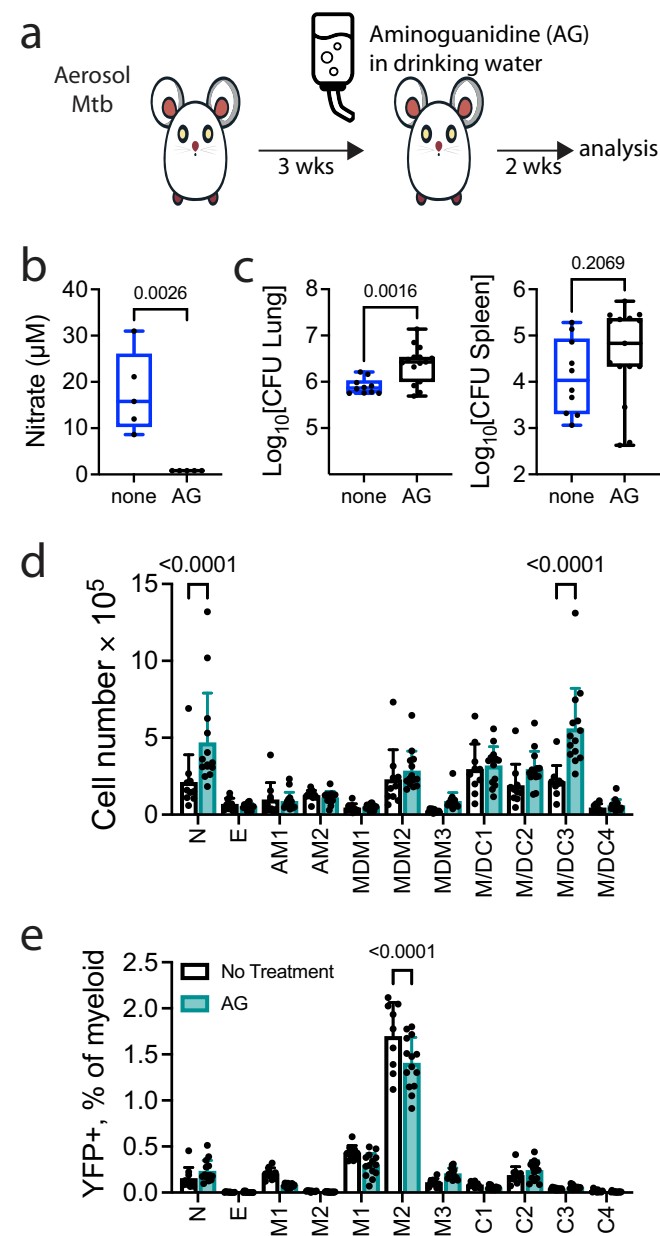

**Fig. 5 | Delayed inhibition of NOS2 has only a small effect on *M. tuberculosis* bacterial burden. a** Experimental scheme for AG treatment. Mice were giving AG in their drinking water starting from week 3 post infection over the course of two weeks. **b** Nitrate measured in lung homogenates of control mice or those treated with aminoguanidine. **c** Lung and spleen CFU from mice treated with or without AG at 5 wpi. **d** Total numbers of myeloid cell populations in the lung at 5 wpi. **e** %YFP⁺ as a fraction of total myeloid cells following AG treatment at 5 wpi. Each point represents an individual mouse (*n* = 5) from two independent experiments. Statistical testing was done by t-test (**b**, **c**) or a two-way ANOVA was performed using Tukey's (**d**) or Dunnett's multiple comparison tests (**e**). P-(two-tailed for **b**, **c**) values are indicated. (**b**, **c**) Box plots indicate median (middle line), 25th, 75th percentile (box) and minimum and maximum (whiskers). **d**, **e** Bars are mean ± SD. Source data are provided as a Source Data file.

would expect that the total number of *M. tuberculosis* infected cells should decrease. Although one sees dramatic reductions in the number of infected AM and neutrophils, the number of infected MDM and other cells did not significantly change (Fig. 6e, left). In fact, because of these changes, CD4 T cells induce a shift in the dominant infected cell type from neutrophils to MDM (Fig. 6e, right).

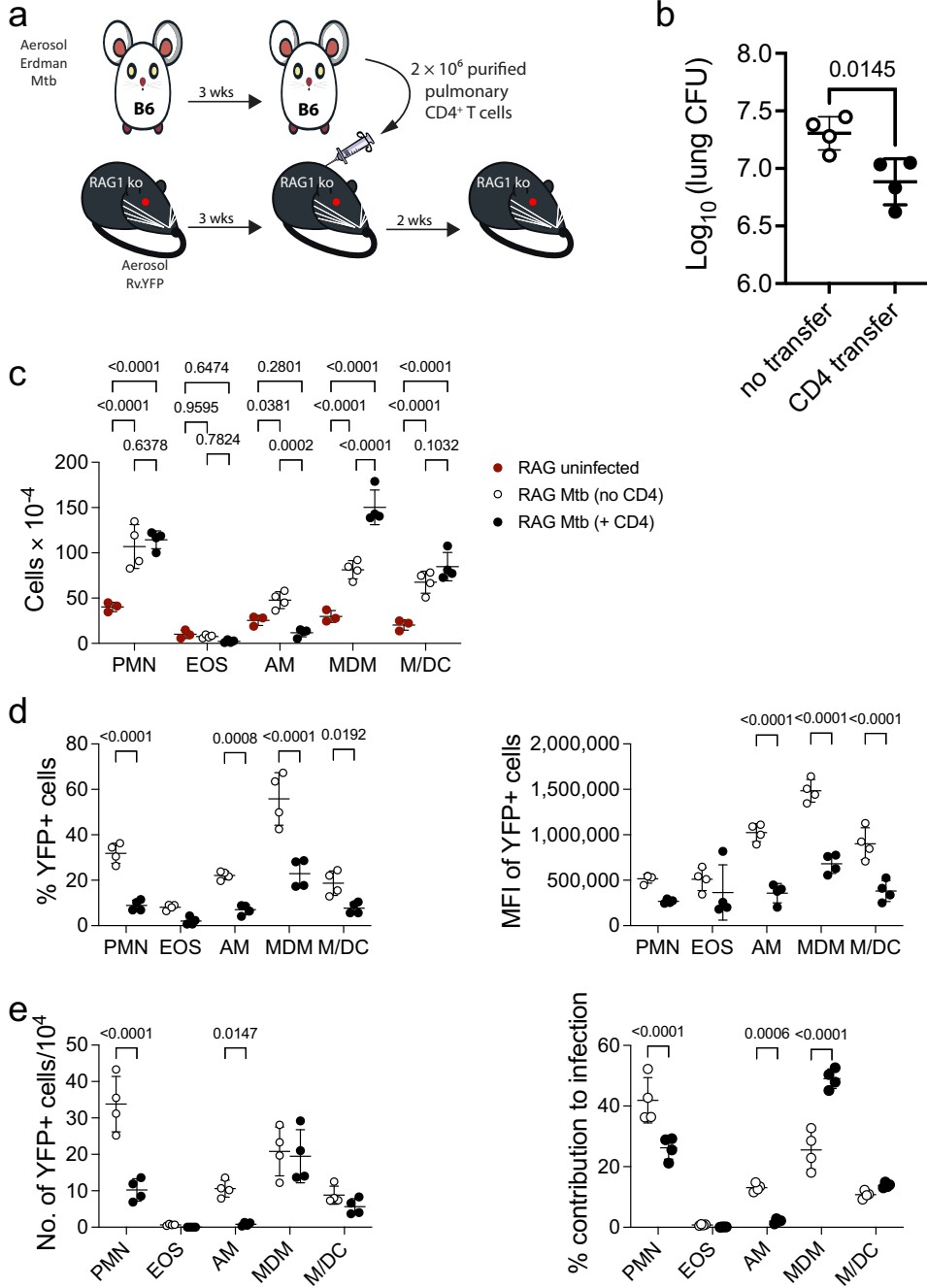

**Fig. 6 | Adoptive transfer of CD4 T cells drive recruitment of MDM to the lung.**
**a** Experimental scheme for CD4 T cell adoptive transfer. RAG1 KO were infected
with Rv.YFP as indicated and rested for 3 weeks. In parallel, C57BL/6 mice were
infected with *M. tuberculosis*, and at 3 wpi, polyclonal CD4 T cells were purified and
transferred i.v. into infected RAG1 KO mice. Cells were isolated from RAG1 KO mice
two weeks post transfer, and bacterial burden was measured. **b** Lung CFU was
determined in RAG1 KO mice that either received CD4 T cells or PBS. **c** Total
numbers of the different myeloid populations. **d** % YFP (left) and the MFI of YFP

(right) within each myeloid population. **e** Total number of YFP+ cells within each
myeloid subset (left) and their fractional contribution to the total number of
infected cells (right). The data (n = 4) is one of three independent experiments.
Statistical analysis was performed using a t-test (**b**), or two-way ANOVA with Šídák's
multiple comparison test (**c–e**). The violin plots show the 25th percentile, median,
and 75th percentile. **b** Bars are mean ± SEM. P (two-tailed for **b**) values are indicated.
Source data are provided as a Source Data file.

## Discussion

The interaction between T cells and infected phagocytes determines
the course of *M. tuberculosis* infection. While T cell immunity is
required to contain infection, it is not sufficient to sterilize mice or the
approximately 10% of people that develop active tuberculosis. In the
mouse TB model, infection of pulmonary DC has been reported, but
using better lineage markers, multiparametric flow cytometry, and
single cell RNASeq, it now appears that CD11c+ macrophages are the

dominant infected macrophage population in the lung[9,11,12]. Given the
inherent diversity of cell types, activation states, and degree of infec-
tion of cells in the lung, we asked whether the impact of T cell
immunity is equally distributed among different myeloid cell types. We
hypothesized that there could exist cellular niches in which *M. tuber-
culosis* is able to persist because T cells are unable to recognize certain
infected cells. Here we report that T cell control of *M. tuberculosis*
infection depends on the type of infected cell. *M. tuberculosis*-infection

of AM is efficiently limited by T cells. In contrast, T cells have only a modest impact on controlling *M. tuberculosis*-infection among MDM.

Using a newly designed flow antibody panel that takes advantage of spectral flow cytometers, we validated our previous results and those of other labs. Although our techniques do not address the tissue localization of the infected cells, at three weeks after low dose aerosol *M. tuberculosis* infection, most bacilli are found within three cell populations: neutrophils, macrophages, and monocyte/DCs. By five weeks, relatively few bacteria reside in DC or monocytes; instead, most *M. tuberculosis* is within macrophages, confirming that macrophages are its major niche. The macrophages are of two types: AM and MDM. With additional markers, the macrophages can be subdivided into 5 populations. A priori, these different macrophage populations could reflect different ontogeny or activation states. For example, significantly more CD11b[+] AM were infected than CD11b[−] AM. As CD11b is upregulated upon AM activation[9,11,12], we speculate that uninfected CD11b[−] AM could occupy uninfected regions of the lung. Similarly, we find that the expression of SiglecF and CD11c among the MDM defined three distinct populations and responded differently to *M. tuberculosis* infection.

CD8 depletion had little or no effect on early *M. tuberculosis* recrudescence in the lung. Similarly, CD4 depletion led to a statistically significant increase in only one of three experiments. In contrast, dual depletion led to a dramatic increase in *M. tuberculosis* burden in the lung. The increase in lung and spleen CFU tracked with the distribution of Rv.YFP in infected cells. An exact correlation between CFU and YFP[+] should not be expected as flow cytometric analysis cannot assess extracellular Mtb or differentiate between live and dead bacilli. Nevertheless, CD4 depletion increased the frequency of YFP[+] neutrophils and macrophages, and the increase of *M. tuberculosis*-infected cells was even greater after dual CD4 and CD8 depletion. This is consistent with our previous observations where the increase in neutrophil recruitment is associated with the loss of Th1 immunity and increased disease severity[33]. Although neutrophils are highly effective at the uptake of Mtb, whether neutrophils are capable of directly killing Mtb remains a contentious issue[34]. However, neutrophils respond poorly to adaptive T cell responses in comparison to macrophages[35]. As such, we believe that the increase in neutrophil burden is due to a loss of containment within the macrophage subsets.

The greater perturbation in the distribution of *M. tuberculosis* among myeloid cells following CD4 but not CD8 depletion suggests that CD4 and CD8 T cell effectors are only partially redundant[16,18]. Alternatively, depletion of CD4 T cells might not be efficiently compensated by CD8 T cells because CD4 T cell help is required to maintain CD8 T cell antibacterial effector function[36]. The large changes in the frequency of infected cells that occurred when both CD4 and CD8 T cells were depleted indicates that CD4 and CD8 T cells have a synergistic role in mediating control of *M. tuberculosis* infection in both AM and MDM subsets.

NOS2 is essential for host resistance to *M. tuberculosis* infection in B6 mice, and mice that lack the NOS2 gene succumb to infection after 4-5 weeks[30]. Induction of nitric oxide (NO) requires two canonical signals: IFNγ and a microbial signal such as LPS[37]. Nitric oxide (NO) production by NOS2-expressing macrophages can kill *M. tuberculosis* in vitro and has led to the paradigm that IFNγ production by CD4 T cells induces NO production by macrophages, leading to control of *M. tuberculosis*[38]. The in vivo role of NO is less certain. NO regulates inflammation during *M. tuberculosis* infection in vivo[32]. We find NOS2 expression by macrophages was diminished only after CD4 depletion. NOS2 expression by macrophages was similar after CD4 depletion or dual CD4 and CD8 depletion, indicating that CD8 T cells had no effect on NOS2 expression and suggesting that CD8 T cells may not rely on the induction of NOS2 to control *M. tuberculosis*.

As CD4 depletion led to reduced NOS2 expression and increased frequency of YFP[+] macrophages, in vivo inhibition of NOS2 should mimic CD4 depletion and lead to a loss of bacterial control. Surprisingly, aminoguanidine treatment of *M. tuberculosis* infected mice had no effect on the *M. tuberculosis* burden within either AM or MDM, despite evidence of NO inhibition. However, we do observe increased recruitment of neutrophils as well as an increase in overall lung CFU, both of which have been previous reported following treatment with aminoguanidine[32]. One possibility is that the length of the treatment wasn't long enough to significantly perturb the intracellular burden to the point where we can detect intracellular changes. Another possibility is the timing of aminoguanidine treatment. A difference between our protocol and previous studies[32,39] is that we treated with aminoguanidine after recruitment of T cell responses to the lung (i.e., starting 3 wpi). The minimal change in *M. tuberculosis* burden following aminoguanidine treatment after three weeks may indicate that NO is no longer crucial for *M. tuberculosis* containment once T cell immunity is generated. It may also point toward an IFNγ-independent effector mechanism for controlling the growth of *M. tuberculosis* within AM and MDM[40–42].

The idea that AM ineffectively restricts *M. tuberculosis* is based on their M2-like nature and improved host control of pulmonary *M. tuberculosis* after their depletion[43]. David Russell's lab elegantly showed that IM (herein referred to as MDM) have a greater intrinsic ability to control *M. tuberculosis* than AM[12]. Thus, AM appear to be a sanctuary for *M. tuberculosis*. Yet, these data focus on early events after infection. Our data provide additional context for what subsequently happens. Early on, neutrophils, macrophages, and monocytes/DCs are infected similarly. However, once immune T cells are recruited to the lung, monocyte-derived macrophages harbor the bulk of *M. tuberculosis* by five weeks after infection. A caveat of our studies is that they focus on relatively early (i.e., 3–5 wpi) interactions between T cells and lung macrophage subsets. Many studies show that early and late T cell responses differ greatly[44]. It will be important to characterize lung macrophage subsets and their interactions with T cells during chronic infection, although this remains technically challenging. Still, it is paradoxical that CD11c[+] MDM are better able to control *M. tuberculosis* infection[12] but are also the major niche for *M. tuberculosis*[9]. Our data shows that the relative permissiveness of macrophage for *M. tuberculosis* is modified by T cells. Initially, AM poorly constrain *M. tuberculosis*, but T cells dramatically improve the ability of AM to control *M. tuberculosis* infection. In the absence of T cells, AM emerge again as a haven for *M. tuberculosis*. In contrast, MDM become a niche where *M. tuberculosis* persists in the long term even in the presence of T cells.

Why would this be the case? We speculate that MDM are already optimally activated following *M. tuberculosis* infection and MDM have an intrinsic capacity to restrict *M. tuberculosis*. While T cells promote MDM control of *M. tuberculosis* infection, the magnitude of this effect is less than for AM. T cells, and T cell factors adds little to the intrinsic ability of MDM to control *M. tuberculosis* infection. Over time, their intrinsic activation could potentially impair their APC function if degradation of *M. tuberculosis* antigens reduced their flow into antigen presentation pathways. Alternatively, we consider the possibility that macrophages with only one or two bacteria could be difficult for T cells to recognize. Interestingly, we observed that T cells reduced the bacillary content of infected cells to a point, and the minimum value was similar for all the different cell types. If the intrinsic antibacterial activity of MDM reduces the intracellular bacterial burden without killing the bacilli, the result could be a macrophage where *M. tuberculosis* persists but can't be recognized by T cells. Finally, there is an additional factor. Transfer of CD4 T cells to RAG1 KO mice leads to reduced CFU and YFP signal compared to un-transferred mice, showing CD4 T cells are critical to protection. However, CD4 T cells also promote the recruitment of macrophages to the lung, presumably through the elaboration of chemokines such as CCL2, CCL5, or CX3CL1[45,46]. Although monocyte and macrophage recruitment to an inflammatory site is generally viewed as a beneficial response, in the

case of tuberculosis, it provides a new crop of macrophages that can be infected by *M. tuberculosis* and are permissive for its survival and replication.

## Methods

### Ethics statement

Studies were conducted using the relevant guidelines and regulations and approved by the Institutional Animal Care and Use Committee at the University of Massachusetts Medical School (UMMS) (Animal Welfare A3306-01), using the recommendations from the Guide for the Care and Use of Laboratory Animals of the National Institutes of Health and the Office of Laboratory Animal Welfare.

**Mice.** 6–8-week-old mice were purchased from Jackson Laboratories (Bar Harbor, ME). RAG1$^{-/-}$ mice on the C57BL6/J background were bred in house. All animals in the study were used in the Animal Medicine Facility at the University of Massachusetts Medical School.

**In vivo infection.** C57BL/6 J were purchased from Jackson Laboratories (Bar Harbor, ME). H37Rv expressing yellow fluorescent protein (YFP) has been previously described in refs. 9,35. The mice were exposed to an aerosolized inoculum of Rv.YFP using a Glas-Col Inhalation Exposure System (Glas-Col LLC, Terre Haute, IN) as described in refs. 36,47. The number of *M. tuberculosis* deposited in the lungs was determined for each experiment and varied ranging between 37 and 90 CFU.

**CD4 and CD8 T cell depletion.** 200 μg of either anti-CD4 (clone GK1.5) or anti-CD8 (clone 2.43) mAb were injected intra-peritoneally biweekly starting on day 21 for 14 days to deplete CD4 and/or CD8 T cells.

**Lung cell preparation.** To isolate total lung leukocytes, lungs were perfused by slowly injecting PBS into right ventricle immediately after mice were killed. The lungs were minced with a gentleMACS dissociator (Miltenyi) and digested (30 min, 37 °C) in 250 U/ml collagenase and 60 U/ml DNase (both from Sigma-Aldrich). Lung cell suspensions were passed through a 70 μm and 40 μm strainers sequentially to remove cell clumps. Lung cells were resuspended in autoMACS running buffer (Miltenyi) that contains BSA, EDTA, and 0.09% sodium azide for subsequent staining.

**Flow cytometry analysis.** Information about the antibodies used for flow cytometry can be found in Supplementary Table 1. Cells were first stained with Live/Dead Fixable NIR Dead cell Stain Kit (ThermoFisher) for 10 min at room temperature (RT), after which cells were stained with 5 ug/ml of anti-mouse CD16/32 mAb (BioXcell) in autoMACS running buffer (Miltenyi) for 10 min at 4 °C. Next, the cells were then stained with a surface antibody cocktail for 20 min at 4 °C. Dead cells were excluded using Live/Dead NIR from Thermofisher (L10119). Hematopoietic cells were first gated on using CD45.2 (clone 104, 1:200), after which a dump channel was used to exclude T cells, B cells, and NK cells for efficient myeloid cell analysis and included anti-Thy1.2 (clone 30H12, 1:200), anti-CD19 (clone 6D5, 1:600), and anti-NK1.1 (clone PK136, 1:100) with BV421. Macrophages were subsequently defined based on a combination of CD11c on Pacific Blue (clone N418, 1:150), SiglecF on BV786 (clone E50–2440, 1:200), Mertk on PE (clone 2B10C42, 1:50), CD64 on PE-Cy7 (clone X54-5/7.1, 1:250), Ly6C on PerCP-Cy5.5 (clone HK1.4, 1:133) and CD11b on Alexafluor 594 (clone: M1/70, 1:250). Cells were permeabilized using Cytofix/Cytoperm fixation/permeabilization kit from BD (554714), after which intracellular iNOS on APC (CXNFT, 1:200) was examined. To inactivate the bacteria, samples were fixed with 1% paraformaldehyde/PBS for 1 h at room temperature and then washed with MACS buffer. Samples were run on either a 4 or 5 laser Cytek Aurora. Erdman (non-fluorescent) infected mice used for unstained control, single stains and YFP-FMO. Autofluorescence (AF) was treated as a fluorescent parameter during unmixing either automatically through the SpectralFlo software v3.1 or by manually deriving an autofluorescence fingerprint (assigned to BV510). After unmixing FMOs were used to define positive and negative populations for each parameter. Flow data were analyzed using FlowJo v10.7.1.

**Adoptive transfer model.** Spleens and lymph nodes from C57BL/6 J mice were mechanically disrupted onto 70 μm strainers using the plungers of 3 mL syringes. CD4 T cells were purified from spleens and lymph nodes using MojoSort CD4 isolation kit and magnet (Biolegend). Purities of cells were determined for each experiment using flow cytometry. 2–5 million CD4 T cells were transferred into RAG1 KO mice before infecting with Rv.YFP.

**Cell sorting.** Cells were sorted using Sony MA900 located in the biosafety level 3 lab in University of Massachusetts Medical School using the v3.1 software. Cells were stained with Zombie violet to exclude dead cells. Target cells were separated 2-way into polypropylene FACS tubes containing 2 ml of FBS. CD45 (clone 104) was used to identify hematopoietic cells, after which Thy1.2 (clone 30H12), CD19 (clone 6D5) and Ly6G (clone 1A8) were used to remove lymphocytes and neutrophils. Macrophages were identified using a combination of CD11c (clone N418), SiglecF (clone E50–2440), Mertk (clone 2B10C42), CD64 (clone X54-5/7.1).

**Aminoguanidine treatment and NO quantification.** Mice were supplied with drinking water containing 2.5% aminoguanidine hemisulfate (A7009-100G Sigma) starting from week 3 post infection. Water was replaced once a week. Nitric oxide levels were quantified in lung homogenates using a Greiss assay (G4410-10G Sigma) as previously described in ref. 32.

**Statistical analysis.** Statistical analysis was performed using Prism 9 (GraphPad). *P*-values were calculated using unpaired two-sided t test, one-way ANOVA, or as indicated in the figure legends. For normally distributed data an ordinary one-way ANOVA was used followed by Tukey's multiple comparisons test. If selected means were compared, Šídák's multiple comparisons test was used. If the data was not normally distributed, Welch's test was used, followed by the Dunnett multiple comparison test. For two-way ANOVA, Bonferroni multiple comparison test was used.

### Reporting summary

Further information on research design is available in the Nature Portfolio Reporting Summary linked to this article.

## Data availability

All data supporting the findings of this study are available within the paper and its supplementary information. Further information and requests for resources and reagents should be directed to and will be fulfilled by the Corresponding author, Samuel Behar (Samuel.Behar@umassmed.edu). Source data are provided with this paper.

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

## Acknowledgements

This work was funded by NIH/NIAID grants to S.M.B.: R01AI106725, R01 AI123286, and P01 AI132130.

## Author contributions
Conceptualization, R.L. and S.M.B.; Methodology, R.L., and J.L.; Investigation, R.L., T.W., T.R., and J.L.; Formal analysis, R.L., T.W., and S.M.B.; Writing & Editing, R.L., T.W., T.R., and J.L., and S.M.B.; Supervision, S.M.B.; Funding Acquisition, S.M.B.

## Competing interests
The authors declare no competing interests.
