## [Peer Review File · Nature Communications]

Heterogeneity in lung macrophage control of *Mycobacterium tuberculosis* is modulated by T cellsREVIEWER COMMENTS

Reviewer #1 (Remarks to the Author):

The work of Lai, et al represents an important advance in our understanding of pulmonary myeloid cells that respond to Mtb infection during a critical phase of the infection process. Advanced spectral cytometry is used to identify important shifts in myeloid subpopulations that occur during the 3-5 weeks of infection that correspond with development of the adaptive response by T cells. Classical T cell subset depletion and adoptive transfer experiments further demonstrate the mechanistic role of T cell-mediated immunity in the myeloid response at a level of subpopulation granularity not previously described. These outcomes are an important contribution to understanding the definitive players in the "changing battlefield" during progression of infection. There are several important outcomes including: 1) There is significant heterogeneity among myeloid subsets and even subpopulations within subset regarding susceptibility to infection, 2) The myeloid cells recruited by T cells become an important reservoir cell that offsets the benefits of T cell help for granulocytes and resident myeloid cells, 3) MDM (IM) populations, and especially the IM1 cells are the predominant infected pool despite highly efficient CD4-mediated activation of NOS2, 4) Myeloid cell iNOS activation is strongly associated with infection and not with apparent effector function since iNOS correlates positively and strongly with infection each cell type and inhibition of iNOS during this disease stage fails to change bacterial burden. T cell subset depletion supports the established paradigm of CD4 cells as the predominant driver of iNOS activation in macrophage and DCs, presumably through IFN- γ roles that are not addressed through cytokine analysis.

A criticism of the work is that some of the interpretations are not fully supported by the data, especially that CD8 T cell depletion differentially affects pulmonary MDM (IM) and T cell depletion has the least effect on MDM (IM) populations. Evidence supporting efficacy of aminoguanidine treatment is also lacking, although this is a well-established approach. The lack of additional markers to delineate monocyte and DCs is a somewhat limiting for a comprehensive analysis of myeloid populations. A clear case that recruitment of MDM (IM) populations due to T cells overwhelms Mtb growth restriction mechanisms is made. The

case that MDM (IM) have limited ability to restrict Mtb-infected MDM (IM) cells, however, is overstated based on the data presented.

Specific comments

Data do not support statements that CD8 are more important for control of MDM (IM) than AM. When looking at the AM vs MDM populations in aggregate, it appears to be more of a trend for CD8+CD4 depletion to effect AM while MDM aggregate responses appear to be primarily a CD4 response.

The need for use of new definitions for macrophage populations as MDM1, MDM2, MDM3 is unclear. How do these differ from interstitial (IM) populations such as IM1, IM2, IM3 etc that have been defined with mostly overlapping phenotype markers?

The pooling of monocyte and DC subpopulations makes it difficult to interpret the findings. Inclusion of a 2-3 additional markers in the panel would have allowed a comprehensive analysis.

Some polish in flow gating and analysis is needed. AF should be defined (Autofluorescence presumably) in methods text and figure legend text for flow cytometry gating of macrophage populations. Cleaner gating for CD11b is needed although results are not likely to change significantly.

The sum of the data indicates that Macrophage populations are more susceptible to infection than other myeloid population in addition to greater expansion between 3-5 wk

The results suggest the pooled monocyte/DC populations are abundant and yet generally more resistant to Mtb at 5 wks independent of T cell-mediated immunity. Having expanded markers to identify the M/DC2 population is needed to determine an important Mtb infected host that appears to respond to T cell pressure

Figure 3 data does not support interpretation that CD8 cells differentially effect infection as % or YFP MFI of MDM (IM) populations. The outcomes of CD4, CD8, and double depletion on outcomes of AM vs MDM are overall fairly similar biologically despite some variations that impact statistical strength

The interpretation that MDM are the least impacted by T cell depletion does not appear to be supported by the data in Fig. 3 and 4 regarding infection and iNOS activation. T cell depletion appears to similarly impact AM and MDM (IM) populations. In contrast, the data in sum (including adoptive transfer) do support the interpretation that CD4-dependent recruitment of MDM as new hosts offsets gains in host restriction by CMI. The argument would be further strengthened by inclusion of data demonstrating reduction in total MDM vs other cell types following CD4 depletion.

Use of IFN-g deficient animals as source of CD4 cells in adoptive transfer experiments would have further closed the loop regarding the insufficient role for CD4-dependent activation of macrophage iNOS through IFN-g. The outcomes also reveal little regarding a basis for the additive/synergistic effects of CD8 cells, except to show that redundant activation of iNOS is unlikely to be the mechanism.

The authors statement that the increased % of infected PMN following depletion of both CD4 and CD8 indicates redundant roles requires clarification, including a caveat for this outcome to simply reflecting overall increases in Mtb proliferation. Related to this, the discussion should be expanded to include indirect and direct outcomes and likely pathways and whether infection vs phagocytosis is occurring in YFP+ PMN. If PMN positivity is due to infection, then adoptive transfer studies suggest that CD4 T cells have the most impact on control of bacterial burden through effects on PMN, and AMs.

The failure of AG to impact cell growth is an interesting observation deserving of expanded discussion. Although AG has been implemented in many studies of active, reactivation, or relapse TB, treatment is usually initiated during early or paucibacillary stages of Mtb infection. The lack of effect during established infection at 3 weeks, combined with the iNOS

outcomes observed, makes an important point regarding relevant thresholds and timing of NO responses.

Reviewer #2 (Remarks to the Author):

The manuscript addresses a fundamental question in TB pathophysiology: how T cells affect growth of Mtb. The answer to this question is important as it has a major impact on TB vaccine design. The authors used the low-dose TB mouse model and implemented multiparameter spectral flow cytometry along with cell depletion and cell transfer studies to pin down roles of CD4⁺ and CD8⁺ T cells in control of Mtb growth in lung macrophage populations. The studies are very well planned and the conclusions are fully supported by the data. The results advance the understanding about early immune events in TB. The authors convincingly showed that the control of Mtb is mutually controlled by CD4 and CD8 T cells. While Mtb replication in AM was largely controlled by CD4, both T cell subsets were needed to control bacilli in MDM. Despite their basal higher (compared to AM) propensity to restrict Mtb growth, MDM appear to be less responsive to T cell activation. T cells were unable to significantly increase the anti-mycobactericidal activity of MDM. The studies show that T cells only marginally contributed to the ability of MDM to limit Mtb replication. The basis of MDM's antimycobacterial features and the modalities to modulate the MDM Mtb niche late in infection remain to be elucidated.

Overall, these observations are timely, very interesting and relevant for immunology of TB. They also provide a basis for further investigations aiming at clarifying why MDM have these features and how MDM's phenotype can be modulated.

Several issues need to be addressed to ensure consistency and comprehensibility. These are presented below:

-The nomenclature of MDM varies across experiments, sometimes pools (panels in Fig. 4 and 6) of various populations are depicted. This should be better clarified and reasoning should be provided.

In the same line, usage of nomenclature should be scientifically sound. It is inappropriate to include IM under MDM as these are ontogenically distinct. They may derive from monocytes

if the niche is emptied, and may have though various fixed or transient phenotypes (DOI: 10.1016/j.it.2020.08.008). This could be discussed.

-The authors state that the MFI is not an accurate way to quantify Mtb load per cell (lines 293-295), but used this parameter selectively. It should be used throughout the investigations as it does provide important information regarding Mtb loads (e.g. Fig. 2, Fig. 4).

-FACS analysis does not provide information on tissue localization. Specifically for effector functions this is an important aspect to consider. It would be helpful to provide IHC analysis of NO or iNOS within tissue, best along staining with Mtb and macrophages upon T cell depletion. Alternatively, this limitation should be indicated.

-Does T cell depletion/transfer alter viability of myeloid cell subsets? With impact on the major subsets containing mycobacteria? T cell derived IFN γ alters neutrophil viability in murine TB, and IFN γ -induced NO macrophage apoptosis in vitro. L/D stain could be at informative.

Does T cell depletion alter the distribution of Mtb+ cells within a respective population? (not % of myeloid cells, but % parent population; e.g. Fig. 4 and Fig. 5). This can be easily evaluated based on acquired FACS data.

What is the explanation for having more AM in RAG mice which received CD4 cells?

-The data showing reduced NO production upon aminoguanidine treatment should be shown.

-It would be good to have evidence for chemokine production in presence of T cells, eventually evaluate abundances following depletion to strengthen the hypothesis that chemokine concentrations are key for MDM recruitment.

-The details about the transfer experiments should be checked. The methodology does not match the figure (Fig. 6) and figure legend details (source of cells, mouse genotype).

-Certain statistical information is missing: was normality checked? And considered for the choice of the statistical tests? Why various tests were used for similar groups and experimental setup, an explanation for the choice of specific statistical test must be provided.

-Details about data presentation (mean, median; SD, SEM, IQR etc) are missing in legends, what is shown?

-Please check the text carefully for comprehensibility, e.g. lines 324, 332-333.

Reviewer #3 (Remarks to the Author):

This study deciphers the capacity of bacterial control by several macrophage populations during infection with *M. tuberculosis*. Most importantly, it reports on the variable dependence of T cells to Mtb control by several types of macrophages. The manuscript dissects and quantitates time course changes on infected macrophage populations (alveolar macrophages or AM versus monocyte derived macrophages or MDMs) during the course of Mtb infection in the lungs. As alveolar macrophages are primary cells infected by Mtb and control infection but do not sterilize with subsequent initiation of an inflammatory response that recruit monocytes to the lungs and activate T cell responses. As result, the monocytes-macrophages arriving to the lungs are also infected and are also unable to sterilize the bacteria. Later on the manuscript describes the important role of CD4 and CD8 T cells recruited to the lungs in controlling infection by Mtb infected AMs and MDM1; MDM2; MDM3. Using T cell depletion, transfer approaches along with KO mice, the authors demonstrate that while CD4 T cells are needed to control infection in AM and MDM1; control of Mtb infected MDM2 requires also CD8 T cells. Overall the study shows that T cells have only modest impact in controlling infection MDM cells. Most of the research approach uses flow cytometry, cell sorting, bacterial burden determination, cell depletion and knockout mice. While there were no concerns about the technical aspects, data analysis and conclusions derived from these studies, this reviewer had minor concerns regarding some terminology used;

1.- The use of term "intracellular infection" (e.g. lines 255, 257 among others) when referring to data obtained from flow cytometry assay is problematic. Flow cytometry cannot differentiate between fluorescent bacilli adhered to cell membrane and intracellular location. One cannot refer to intracellular infection when using simply flow cytometry. No studies in this manuscript search for intracellular bacilli in any way. One way to overcome this limitation would be to do confocal imaging of cell sorted macrophages to determine the percentage of macrophages bearing intracellular bacilli (Fig 2D) in the macrophage sorted population

2.- Also confusing was the term "intracellular infection" in line 496 (among others) when referring to data obtained by CFU enumeration of lungs and spleens. Similar to flow cytometry, when enumerating CFU using lung homogenates, it is not possible to discriminate between intracellular and extracellular bacilli.

3.- use of (n=5/group) through manuscript. In statistics n=5 refers to 5 in a group. So n=5 is sufficient to refer to mice per group.

Otherwise , this is a study of great importance to the field of tuberculosis

REVIEWER COMMENTS

Reviewer #1 (Remarks to the Author):

The work of Lai, et al represents an important advance in our understanding of pulmonary myeloid cells that respond to Mtb infection during a critical phase of the infection process. Advanced spectral cytometry is used to identify important shifts in myeloid subpopulations that occur during the 3-5 weeks of infection that correspond with development of the adaptive response by T cells. Classical T cell subset depletion and adoptive transfer experiments further demonstrate the mechanistic role of T cell-mediated immunity in the myeloid response at a level of subpopulation granularity not previously described. These outcomes are an important contribution to understanding the definitive players in the "changing battlefield" during progression of infection. There are several important outcomes including: 1) There is significant heterogeneity among myeloid subsets and even subpopulations within subset regarding susceptibility to infection, 2) The myeloid cells recruited by T cells become an important reservoir cell that offsets the benefits of T cell help for granulocytes and resident myeloid cells, 3) MDM (IM) populations, and especially the IM1 cells are the predominant infected pool despite highly efficient CD4-mediated activation of NOS2, 4) Myeloid cell iNOS activation is strongly associated with infection and not with apparent effector function since iNOS correlates positively and strongly with infection each cell type and inhibition of iNOS during this disease stage fails to change bacterial burden. T cell subset depletion supports the established paradigm of CD4 cells as the predominant driver of iNOS activation in macrophage and DCs, presumably through IFN- γ roles that are not addressed through cytokine analysis.

We thank the reviewer for the support of our work.

A criticism of the work is that some of the interpretations are not fully supported by the data, especially that CD8 T cell depletion differentially affects pulmonary MDM (IM) and T cell depletion has the least effect on MDM (IM) populations. Evidence supporting efficacy of aminoguanidine treatment is also lacking, although this is a well-established approach. The lack of additional markers to delineate monocyte and DCs is a somewhat limiting for a comprehensive analysis of myeloid populations. A clear case that recruitment of MDM (IM) populations due to T cells overwhelms Mtb growth restriction mechanisms is made. The case that MDM (IM) have limited ability to restrict Mtb-infected MDM (IM) cells, however, is overstated based on the data presented.

We have addressed the specific comments below.

Specific comments

1. Data do not support statements that CD8 are more important for control of MDM (IM) than AM. When looking at the AM vs MDM populations in aggregate, it appears to be more of a trend for CD8+CD4 depletion to effect AM while MDM aggregate responses appear to be primarily a CD4 response.

We based our statements on the data for the AM1 and MDM1 presented in Fig.3c. After re-examining the data in this study, we agree with the reviewer that the difference is small. We have removed lines 303-310 in the original discussion, which the reviewer is referring to.

2. The need for use of new definitions for macrophage populations as MDM1, MDM2, MDM3 is

unclear. How do these differ from interstitial (IM) populations such as IM1, IM2, IM3 etc that have been defined with mostly overlapping phenotype markers?

The reviewer raises the important issue of nomenclature. First, let's acknowledge that the literature is full of different definitions of macrophage subsets. Second, we struggle to align our definitions with what is used by other investigators in the Mtb field (for example, the work from the labs of David Russell^{1,2,3}, Joel Ernst^{4,5}, or Alissa Rothchild^{6,7}) as well as that used by the leading labs that study the ontogeny of macrophages or lung macrophages.

We originally referred to the dominant non-AM macrophage population as CD11c^{Hi} monocyte derived cells (MDC) based largely on the work of Martin Williams^{8,9}. These cells have also been referred to as RM (recruited macrophages – but a confusing abbreviation because it could be resident macrophages) or IM (also confusing because sometimes it refers to interstitial macrophages and other times, inflammatory macrophages). In our current work, we have changed our own nomenclature for two reasons. First, not all these cells are CD11c^{Hi} – their CD11c expression can be variable. Second, MDC is too much like the abbreviation mDC (“myeloid DC”). Instead, we now refer to these as MDM for monocyte-derived macrophages, which denotes both their ontological derivation and the fact that they are macrophages. This is described in lines 121-124.

Why not call them IM? Although we agree that they are similar to the IM as defined in the substantial body of work from David Russell, we need to balance this with the substantial body of work on IM from uninfected mice^{10, 11, 12, 13, 14}. It is not clear whether the IM1, IM2, and IM3 found in normal lung are the same cells that are recruited to the lung during Mtb. This becomes a fundamental question as to the origins of IMs in the normal lung, a question that is quite different from the one that we are asking. We are quite certain that the macrophages that we are studying are recruited to the lung and derived from monocytes, based on our own work¹⁵ and that from the Ernst lab⁴. Thus, we would argue that the recruited macrophages in the lungs of Mtb-infected mice are not derived from lung resident interstitial macrophages, but it is possible that interstitial macrophages are derived from monocytes. Regardless, this is not the question we are addressing in this paper. While a unified nomenclature would be terrific, and no doubt will probably emerge in the next couple of years based on more single cell analysis of the heterogeneity and ontogeny of macrophages, to force a single nomenclature on investigators is premature. Clearly, many investigators in the field agree since there is still a variety of definitions being used^{1, 6, 16}.

3. The pooling of monocyte and DC subpopulations makes it difficult to interpret the findings. Inclusion of a 2-3 additional markers in the panel would have allowed a comprehensive analysis.

We agree that a more comprehensive flow panel would have facilitated more definitive identification of the different populations of monocyte and DCs. We did include additional markers in our panel and have included this data in a new supplemental figure (see Figure S2). In Fig.2 (and Fig.S1) we subset these non-macrophages (i.e., Mertk⁻CD64^{lo}) by CD11c and Ly6c into four populations. The additional antibodies in our panel show that M/DC1 and MDC/2 (both CD11c⁺) also express CD26, which is expressed by nearly all murine cDC (10.1016/j.immuni.2020.04.005). This together with their expression of high levels of MHCII make it likely that M/DC1 and MDC/2 are cDC. In contrast, the high levels of Ly6c and CD11b expressed by the CD11c⁻ M/DC3 cells, is typical of monocytes. M/DC4 expresses low levels of CD11c, Ly6c, and the four additional markers, making it difficult to know the identity of this small population of cells. Importantly, our question focuses on how T cell pressure affects the Mtb containment by infected cells. As our analyses show that these cells (with possible exception of M/DC2 at 3wpi) are minimally infected,

our subsequent analysis focused on resolving the different macrophage populations. Also, since these populations are likely to be mixed, even at this level of resolution, we believe referring to them as mixed populations (e.g., M/DC3) is appropriate. We have added these descriptions to the revised manuscript starting on line 129.

4. Some polish in flow gating and analysis is needed. AF should be defined (Autofluorescence presumably) in methods text and figure legend text for flow cytometry gating of macrophage populations. Cleaner gating for CD11b is needed although results are not likely to change significantly.

Autofluorescence is now defined in the legend for Figure 2. In addition, we have re-examined our gates for CD11b on our alveolar macrophages. AM are normally negative for CD11b, although they have been shown to upregulate this marker upon inflammation¹⁷ is the main reason we separated AM1 and AM2. All our gates (including CD11b) were always cross-referenced to FMO controls so we feel confident in the placement of our gates.

5. The sum of the data indicates that Macrophage populations are more susceptible to infection than other myeloid population in addition to greater expansion between 3-5 wk

We thank the reviewer for pointing this out and have added a statement to describe this observation (line 286-287).

6. The results suggest the pooled monocyte/DC populations are abundant and yet generally more resistant to Mtb at 5 wks independent of T cell-mediated immunity. Having expanded markers to identify the M/DC2 population is needed to determine an important Mtb infected host that appears to respond to T cell pressure

The reviewer makes a good point. The reduction in Mtb-infected M/DC2s between 3 and 5 wpi is dramatic and indicates that this cell can largely eliminate infection. Nor do we believe that our data indicates that they respond to T cell pressure, as even with complete T cell depletion, there is only a nonsignificant increase in their frequency that is infected. However, as described above, even with the additional markers in our panel, we are not able to definitively identify the cell type other than to surmise that it is predominantly monocyte-derived DCs. While this is an interesting question for further study, we are unable to say anything definitive about this population.

7. Figure 3 data does not support interpretation that CD8 cells differentially effect infection as % or YFP MFI of MDM (IM) populations. The outcomes of CD4, CD8, and double depletion on outcomes of AM vs MDM are overall fairly similar biologically despite some variations that impact statistical strength

As indicated in our response to comment 1, we agree that CD8 T cells do not differentially affect infection; the lines in the discussion (which we have removed) is the only place where we made this claim. With respect to the outcomes of T cell depletion on AM vs. MDM, we believe that our data convincingly shows that there are important differences between AM and MDM. For example, compares AM1 vs. MDM2, the latter which is the major cell reservoir for Mtb. In the absence of T cells (i.e., the double depletion condition), about 26% of each population is infected. However, in the presence of T cells (i.e., no depletion), only 2% of AM1 are infected while 10% of MDM2 are infected. Furthermore, T cells are responsible for a 3-fold reduction in Rv.YFP MFI in AM1; yet there is no significant reduction of Rv.YFP MFI in MDM2. This is more than just an issue of statistical strength.

8. The interpretation that MDM are the least impacted by T cell depletion does not appear to be supported by the data in Fig. 3 and 4 regarding infection and iNOS activation. T cell depletion appears to similarly impact AM and MDM (IM) populations. In contrast, the data in sum (including adoptive transfer) do support the interpretation that CD4-dependent recruitment of MDM as new hosts offsets gains in host restriction by CMI. The argument would be further strengthened by inclusion of data demonstrating reduction in total MDM vs other cell types following CD4 depletion.

There are three different MDM subsets, and we may not have emphasized that the most important subset is the MDM2 subset. Of the three subsets, they are the most abundant and the most highly infected. Also, by 5 wpi, most Mtb is associated with MDM2 (see Fig. 2c). In Fig.3b, the “YFP, % of subset” represents the percentage of each cell subset that is infected by Mtb. The number above the double depletion represents fold change increase in the percentage of infected cell over undepleted. As the proportion of AM increase significantly higher than the MDM when T cells are depleted, we concluded that the MDM are less impacted by the loss of T cell pressure than AM. This is most clearly seen in Fig.3a, which shows that flow data for a representative mouse from each condition. In the absence of T cells, ~22% of AM1 and MDM2 are YFP+. In the presence of T cells, only 3% of AM1 are infected while 15% of MDM2 are infected. This represents a 95% reduction for the AM1 but only a 32% reduction for the MDM2.

In the results section describing NOS2 expression, we do not claim that MDM are differentially impacted by T cell depletion. We did observe that CD8 depletion did paradoxically increase the frequency of AM1 expressing NOS2. This was a small difference and we have removed the sentence reporting this result. Finally, we have altered Figure 4e and the relevant text (lines 222-224) so it now displays the percentages of NOS2 expression by infected AM1 and MDM2, the highly infected populations. We hope that this change makes our message clearer.

With respect to our argument that CD4 T cells are important in recruitment of MDM, we have included data from T cell depletion experiments showing a reduction in percentage and absolute numbers of MDM when CD4 T cells are eliminated (see Line 177 and new Figure S3).

9. Use of IFN-g deficient animals as source of CD4 cells in adoptive transfer experiments would have further closed the loop regarding the insufficient role for CD4-dependent activation of macrophage iNOS through IFN-g. The outcomes also reveal little regarding a basis for the additive/synergistic effects of CD8 cells, except to show that redundant activation of iNOS is unlikely to be the mechanism.

We do not have data using IFN γ ko CD4 T cells in our adoptive transfer model. As we understand the reviewer’s comment, the expected result for such an experiment is that there would still be persistent NOS2 expression in macrophages. We believe that this would be “negative” data and not provide additional insight into the mechanism of CD4-independent expression of NOS2. Also, NOS2 expression is not dependent on CD4 T cells based on prior studies that have used TCR $\alpha\beta$ or MHCII knockout mice and shown that NOS2 expression persists as demonstrated by immunohistochemistry and transcriptional analysis^{18, 19, 20}. Importantly, in these studies, as in ours, there are other sources of IFN γ (NK cells, CD4⁻CD8⁻ T cells). Also, there are many other stimuli that induce NOS2 such as type I IFN^{21, 22} and NOS2 can be induced in IFN γ R ko mice^{23, 24}.

10. The authors statement that the increased % of infected PMN following depletion of both CD4 and CD8 indicates redundant roles requires clarification, including a caveat for this

outcome to simply reflecting overall increases in Mtb proliferation. Related to this, the discussion should be expanded to include indirect and direct outcomes and likely pathways and whether infection vs phagocytosis is occurring in YFP+ PMN. If PMN positivity is due to infection, then adoptive transfer studies suggest that CD4 T cells have the most impact on control of bacterial burden through effects on PMN, and AMs.

We have previously demonstrated that the increase in neutrophil recruitment is due to a loss of Th1 immunity²⁵, which is consistent with our observations following T cell depletion. However, a recent study from Lovewell et al have demonstrated that granulocytes poorly respond to adaptive immunity in comparison to infected macrophages²⁶. Given this observation, we believe that the increase in neutrophil burden is a consequence of a loss of containment in the macrophage subset rather than a direct impact on neutrophil due to the loss of T cells. We have updated our discussion to include these points (line 299 to 307).

11. The failure of AG to impact cell growth is an interesting observation deserving of expanded discussion. Although AG has been implemented in many studies of active, reactivation, or relapse TB, treatment is usually initiated during early or paucibacillary stages of Mtb infection. The lack of effect during established infection at 3 weeks, combined with the iNOS outcomes observed, makes an important point regarding relevant thresholds and timing of NO responses.

We agree and thank the reviewer for highlighting the inability of AG to impact Mtb growth during this critical time period after the arrival of T cell responses. We have updated our discussion to include these points (line 333 to 338).

Reviewer #2 (Remarks to the Author):

The manuscript addresses a fundamental question in TB pathophysiology: how T cells affect growth of Mtb. The answer to this question is important as it has a major impact on TB vaccine design. The authors used the low-dose TB mouse model and implemented multiparameter spectral flow cytometry along with cell depletion and cell transfer studies to pin down roles of CD4+ and CD8+ T cells in control of Mtb growth in lung macrophage populations. The studies are very well planned and the conclusions are fully supported by the data. The results advance the understanding about early immune events in TB. The authors convincingly showed that the control of Mtb is mutually controlled by CD4 and CD8 T cells. While Mtb replication in AM was largely controlled by CD4, both T cell subsets were needed to control bacilli in MDM. Despite their basal higher (compared to AM) propensity to restrict Mtb growth, MDM appear to be less responsive to T cell activation. T cells were unable to significantly increase anti-mycobactericidal activity of MDM. The studies show that T cells only marginally contributed to ability of MDM to limit Mtb replication. The basis of MDM's antimycobacterial features and the modalities to modulate the MDM Mtb niche late in infection remain to be elucidated. Overall, these observations are timely, very interesting and relevant for immunology of TB. They also provide a basis for further investigations aiming at clarifying why MDM have these features and how MDM's phenotype can be modulated.

We thank the reviewer for the support of our work.

Several issues need to be addressed to ensure consistency and comprehensibility. These are presented below:

1. The nomenclature of MDM varies across experiments, sometimes pools (panels in Fig. 4 and 6) of various populations are depicted. This should be better clarified and reasoning should be

provided. In the same line, usage of nomenclature should be scientifically sound. It is inappropriate to include IM under MDM as these are ontogenically distinct. They may derive from monocytes if the niche is emptied, and may have though various fixed or transient phenotypes (DOI: 10.1016/j.it.2020.08.008). This could be discussed.

We agree with the reviewer and have removed the pools from Fig.4. We agree that lung IM should not be confused with MDM. In particular, we specifically avoid using the IM nomenclature exactly because we believe that they differ ontologically, as suggested by the reviewer. See our response to Reviewer 1, point 3 and 8, for more details and a description where in the manuscript our reasoning is provided.

2. The authors state that the MFI is not an accurate way to quantify Mtb load per cell (lines 293-295), but used this parameter selectively. It should be used throughout the investigations as it does provide important information regarding Mtb loads (e.g. Fig. 2, Fig. 4).

We believe that the reviewer misunderstood or misinterpreted our meaning. We agree that the MFI of Rv.YFP is an important metric for quantifying Mtb load by flow cytometry. Indeed, shown correlations between intracellular Mtb and YFP in previous publications^{27, 28}. However, there are limitations. We have edited line 298 in the discussion and removed the phrase "...only poorly assess the number of bacilli/cell." We did not intend this sentence to be dismissive of the value of MFI. In Figure 2, we have flow sorted and quantified the number of intracellular bacteria within MDM and AM by CFU assay, which we believe supports the correlation between MFI and CFU.

3. FACS analysis does not provide information on tissue localization. Specifically for effector functions this is an important aspect to consider. It would be helpful to provide IHC analysis of NO or iNOS within tissue, best along staining with Mtb and macrophages upon T cell depletion. Alternatively, this limitation should be indicated.

We acknowledge that the techniques used in this study fails to address the spatial localization of the infected cells and have included a statement in our discussion to address this (line 282-283).

4. Does T cell depletion/transfer alter viability of myeloid cell subsets? With impact on the major subsets containing mycobacteria? T cell derived IFN γ alters neutrophil viability in murine TB, and IFN γ -induced NO macrophage apoptosis in vitro. L/D stain could be informative. Does T cell depletion alter the distribution of Mtb+ cells within a respective population? (not % of myeloid cells, but % parent population; e.g. Fig. 4 and Fig. 5). This can be easily evaluated based on acquired FACS data.

We have included viability data for the macrophage subsets in a new supplemental figure (see Figure S3). Line 171-172

5. What is the explanation for having more AM in RAG mice which received CD4 cells?

We believe that the reviewer is referring to our data in Figure 6C where we examined the number of myeloid cells in Mtb infected RAG^{-/-} mice, with or without CD4 T cells. However, while we see a significant increase in the number of MDM in mice that received CD4 T cells, we do not see a significant increase in any other population including AM. Maybe the reviewer is referring to the increase in AM following infection of RAG mice with Mtb? We have amended our discussion to increase the clarity of this data. Line 252.

6. The data showing reduced NO production upon aminoguanidine treatment should be shown.

We have now included data showing reduced NO production in the lungs of aminoguanidine treated mice in a revised figure 5 (see Fig.5b).

7. It would be good to have evidence for chemokine production in presence of T cells, eventually evaluate abundances following depletion to strengthen the hypothesis that chemokine concentrations are key for MDM recruitment.

We do not have data looking at chemokine levels in the presence or absence of T cells. However, it is well documented that CD4 T cells influence the production of chemokines by infected macrophages during TB. We have updated our references to support this statement ^{29, 30} (see line 272)

8. The details about the transfer experiments should be checked. The methodology does not match the figure (Fig. 6) and figure legend details (source of cells, mouse genotype).

We thank the reviewer for pointing this out and have corrected the inconsistencies in the methods.

9. Certain statistical information is missing: was normality checked? And considered for the choice of the statistical tests? Why various tests were used for similar groups and experimental setup, an explanation for the choice of specific statistical test must be provided.

We apologize for any omissions and have revised the figure legends and the methods to provide more information about the statistical tests that were used and the rationale for their choice.

10. Details about data presentation (mean, median; SD, SEM, IQR etc) are missing in legends, what is shown?

We have revised the figure legends to include the necessary information to interpret the data in the graphs.

11. Please check the text carefully for comprehensibility, e.g. lines 324, 332-333.

Thank you – we have reread the text and correct several typos.

Reviewer #3 (Remarks to the Author):

This study deciphers the capacity of bacterial control by several macrophage populations during infection with *M. tuberculosis*. Most importantly, it reports on the variable dependance of T cells to Mtb control by several types of macrophages. The manuscript dissects and quantitates time course changes on infected macrophages populations (alveolar macrophages or AM versus monocyte derived macrophages or MDMs) during the course of Mtb infection in the lungs. As alveolar macrophages are primary cells infected by Mtb and control infection but do not sterilize with subsequent initiation of an inflammatory response that recruit monocytes to the lungs and activate T cell responses. As result, the monocytes-macrophages arriving to the lungs are also infected and are also unable to sterilize the bacteria. Later on the manuscript describes the important role of CD4 and CD8 T cells recruited to the lungs in controlling infection by Mtb infected AMs and MDM1; MDM2; MDM3. Using T cell depletion, transfer approaches along with KO mice, the authors demonstrate that while CD4 T cells are needed to control infection in AM and MDM1; control of Mtb infected MDM2 requires also CD8 T cells. Overall the study shows

that T cells have only modest impact in controlling infection MDM cells. Most of the research approach uses flow cytometry, cell sorting, bacterial burden determination, cell depletion and knockout mice. While there were no concerns about the technical aspects, data analysis and conclusions derived from these studies, this reviewer had minor concerns regarding some terminology used;

We thank the reviewer for the support of our work.

1.- The use of term "intracellular infection" (e.g. lines 255, 257 among others) when referring to data obtained from flow cytometry assay is problematic. Flow cytometry cannot differentiate between fluorescent bacilli adhered to cell membrane and intracellular location. One cannot refer to intracellular infection when using simply flow cytometry. No studies in this manuscript search for intracellular bacilli in any way. One way to overcome this limitation would be to do confocal imaging of cell sorted macrophages to determine the percentage of macrophages bearing intracellular bacilli (Fig 2D) in the macrophage sorted population

The reviewer is correct that flow cytometry cannot differentiate between intracellular bacilli and "cell surface" bacteria. In my 20+ years in this field, I have seen this issue raised many times concerning the location of Mtb after in vitro infections of macrophages and more recently for flow cytometric analysis of lung cells from infected mice. In my lab's experience and my recollection of the results of other investigators, Mtb is always intracellular. I have always assumed that this is because once adhered to the cell membrane, phagocytosis proceeds quickly. I admit that we have less experience with the intracellular location of Rv.YFP after isolation of cells from infected lung. However, we have previously published confocal images of macrophages similarly infected and analyzed showing the correlation between the MFI of YFP and the number of bacilli (Fig.2 from Lee et al, CD11c^{hi} monocyte-derived macrophages are a major cellular compartment infected by Mycobacterium tuberculosis. PLoS Pathog. 2020 doi: 10.1371/journal.ppat.1008621. PMID: PMC7319360)²⁷. However, in deference to the reviewer, we have changed most of the references to "intracellular infection" to "*M. tuberculosis* bacterial burden," (i.e., original lines 255, 257, 274, 304, 309, 323, 496).

2.- Also confusing was the term "intracellular infection" in line 496 (among others) when referring to data obtained by CFU enumeration of lungs and spleens. Similar to flow cytometry, when enumerating CFU using lung homogenates, it is not possible to discriminate between intracellular and extracellular bacilli.

The reviewer is correct that CFU enumeration cannot discriminate between intracellular and extracellular bacilli and we have changed "intracellular infection" to "*M. tuberculosis* bacterial burden."

3.- use of (n=5/group) through manuscript. In statistics n=5 refers to 5 in a group. So n=5 is sufficient to refer to mice per group.

We thank the reviewer for pointing this out and have simplified our description as recommended.

Otherwise, this is a study of great importance to the field of tuberculosis

References

1. Pisu, D. *et al.* Single cell analysis of *M. tuberculosis* phenotype and macrophage lineages in the infected lung. *J Exp Med* **218** (2021).
2. Pisu, D., Huang, L., Grenier, J.K. & Russell, D.G. Dual RNA-Seq of Mtb-Infected Macrophages In Vivo Reveals Ontologically Distinct Host-Pathogen Interactions. *Cell Rep* **30**, 335-350 e334 (2020).
3. Huang, L., Nazarova, E.V., Tan, S., Liu, Y. & Russell, D.G. Growth of *Mycobacterium tuberculosis* in vivo segregates with host macrophage metabolism and ontogeny. *J Exp Med* (2018).
4. Norris, B.A. & Ernst, J.D. Mononuclear cell dynamics in *M. tuberculosis* infection provide opportunities for therapeutic intervention. *PLoS Pathog* **14**, e1007154 (2018).
5. Srivastava, S., Ernst, J.D. & Desvignes, L. Beyond macrophages: the diversity of mononuclear cells in tuberculosis. *Immunol. Rev.* **262**, 179-192 (2014).
6. Mai, D. *et al.* Exposure to *Mycobacterium* remodels alveolar macrophages and the early innate response to *Mycobacterium tuberculosis* infection. *PLoS Pathog* **20**, e1011871 (2024).
7. Duffy, F.J. *et al.* A contained *Mycobacterium tuberculosis* mouse infection model predicts active disease and containment in humans. *J Infect Dis* (2021).
8. Guilliams, M. & van de Laar, L. A Hitchhiker's Guide to Myeloid Cell Subsets: Practical Implementation of a Novel Mononuclear Phagocyte Classification System. *Front Immunol* **6**, 406 (2015).
9. Guilliams, M., Mildner, A. & Yona, S. Developmental and Functional Heterogeneity of Monocytes. *Immunity* **49**, 595-613 (2018).
10. Aegerter, H., Lambrecht, B.N. & Jakubzick, C.V. Biology of lung macrophages in health and disease. *Immunity* **55**, 1564-1580 (2022).
11. Schyns, J. *et al.* Non-classical tissue monocytes and two functionally distinct populations of interstitial macrophages populate the mouse lung. *Nat Commun* **10**, 3964 (2019).
12. Chakarov, S. *et al.* Two distinct interstitial macrophage populations coexist across tissues in specific subtissular niches. *Science* **363** (2019).
13. Liegeois, M., Legrand, C., Desmet, C.J., Marichal, T. & Bureau, F. The interstitial macrophage: A long-neglected piece in the puzzle of lung immunity. *Cell Immunol* **330**, 91-96 (2018).
14. Schyns, J., Bureau, F. & Marichal, T. Lung Interstitial Macrophages: Past, Present, and Future. *J Immunol Res* **2018**, 5160794 (2018).

15. Skold, M. & Behar, S.M. Tuberculosis triggers a tissue-dependent program of differentiation and acquisition of effector functions by circulating monocytes. *J Immunol* **181**, 6349-6360 (2008).
16. Zheng, W. *et al.* Mycobacterium tuberculosis resides in lysosome-poor monocyte-derived lung cells during chronic infection. *bioRxiv* (2023).
17. Duan, M. *et al.* CD11b immunophenotyping identifies inflammatory profiles in the mouse and human lungs. *Mucosal Immunol* **9**, 550-563 (2016).
18. Caruso, A.M. *et al.* Mice deficient in CD4 T cells have only transiently diminished levels of IFN-gamma, yet succumb to tuberculosis. *J. Immunol.* **162**, 5407-5416 (1999).
19. Scanga, C.A. *et al.* Depletion of CD4(+) T cells causes reactivation of murine persistent tuberculosis despite continued expression of interferon gamma and nitric oxide synthase 2. *The Journal of experimental medicine* **192**, 347-358 (2000).
20. Mogue, T., Goodrich, M.E., Ryan, L., LaCourse, R. & North, R.J. The relative importance of T cell subsets in immunity and immunopathology of airborne Mycobacterium tuberculosis infection in mice. *The Journal of experimental medicine* **193**, 271-280 (2001).
21. Farlik, M. *et al.* Nonconventional initiation complex assembly by STAT and NF-kappaB transcription factors regulates nitric oxide synthase expression. *Immunity* **33**, 25-34 (2010).
22. Utaisincharoen, P. *et al.* Induction of iNOS expression and antimicrobial activity by interferon (IFN)-beta is distinct from IFN-gamma in Burkholderia pseudomallei-infected mouse macrophages. *Clin Exp Immunol* **136**, 277-283 (2004).
23. Kamijo, R. *et al.* Generation of nitric oxide and clearance of interferon-gamma after BCG infection are impaired in mice that lack the interferon-gamma receptor. *J Inflamm* **46**, 23-31 (1995).
24. Kamijo, R. *et al.* Generation of nitric oxide and induction of major histocompatibility complex class II antigen in macrophages from mice lacking the interferon gamma receptor. *Proc Natl Acad Sci U S A* **90**, 6626-6630 (1993).
25. Nandi, B. & Behar, S.M. Regulation of neutrophils by interferon-gamma limits lung inflammation during tuberculosis infection. *J Exp Med* **208**, 2251-2262 (2011).
26. Lovewell, R.R., Baer, C.E., Mishra, B.B., Smith, C.M. & Sasseti, C.M. Granulocytes act as a niche for Mycobacterium tuberculosis growth. *Mucosal Immunol* (2020).
27. Lee, J. *et al.* CD11c^{hi} monocyte-derived macrophages are a major cellular compartment infected by Mycobacterium tuberculosis. *PLoS Pathog* **16**, e1008621 (2020).
28. Mott, D. *et al.* High Bacillary Burden and the ESX-1 Type VII Secretion System Promote MHC Class I Presentation by Mycobacterium tuberculosis-Infected Macrophages to CD8 T Cells. *J Immunol* **210**, 1531-1542 (2023).

29. Yao, Y. *et al.* Induction of Autonomous Memory Alveolar Macrophages Requires T Cell Help and Is Critical to Trained Immunity. *Cell* **175**, 1634-1650 e1617 (2018).
30. Hoft, S.G. *et al.* The Rate of CD4 T Cell Entry into the Lungs during Mycobacterium tuberculosis Infection Is Determined by Partial and Opposing Effects of Multiple Chemokine Receptors. *Infect Immun* **87** (2019).

REVIEWERS' COMMENTS

Reviewer #1 (Remarks to the Author):

I am satisfied with the revision which addresses my previous concerns. There is a very minor editorial issue with a revised statement in line 333-338, missing word?

Reviewer #2 (Remarks to the Author):

Comments have been adequately addressed.

Reviewer #3 (Remarks to the Author):

The authors addressed previous critiques .

REVIEWER COMMENTS

Reviewer #1 (Remarks to the Author):

I am satisfied with the revision which addresses my previous concerns. There is a very minor editorial issue with a revised statement in line 333-338, missing word?

We thank the reviewer for their support and have corrected the typo.

References

1. Pisu, D. *et al.* Single cell analysis of *M. tuberculosis* phenotype and macrophage lineages in the infected lung. *J Exp Med* **218** (2021).
2. Pisu, D., Huang, L., Grenier, J.K. & Russell, D.G. Dual RNA-Seq of Mtb-Infected Macrophages In Vivo Reveals Ontologically Distinct Host-Pathogen Interactions. *Cell Rep* **30**, 335-350 e334 (2020).
3. Huang, L., Nazarova, E.V., Tan, S., Liu, Y. & Russell, D.G. Growth of Mycobacterium tuberculosis in vivo segregates with host macrophage metabolism and ontogeny. *J Exp Med* (2018).
4. Norris, B.A. & Ernst, J.D. Mononuclear cell dynamics in *M. tuberculosis* infection provide opportunities for therapeutic intervention. *PLoS Pathog* **14**, e1007154 (2018).
5. Srivastava, S., Ernst, J.D. & Desvignes, L. Beyond macrophages: the diversity of mononuclear cells in tuberculosis. *Immunol. Rev.* **262**, 179-192 (2014).
6. Mai, D. *et al.* Exposure to Mycobacterium remodels alveolar macrophages and the early innate response to Mycobacterium tuberculosis infection. *PLoS Pathog* **20**, e1011871 (2024).
7. Duffy, F.J. *et al.* A contained Mycobacterium tuberculosis mouse infection model predicts active disease and containment in humans. *J Infect Dis* (2021).
8. Guilliams, M. & van de Laar, L. A Hitchhiker's Guide to Myeloid Cell Subsets: Practical Implementation of a Novel Mononuclear Phagocyte Classification System. *Front Immunol* **6**, 406 (2015).
9. Guilliams, M., Mildner, A. & Yona, S. Developmental and Functional Heterogeneity of Monocytes. *Immunity* **49**, 595-613 (2018).
10. Aegerter, H., Lambrecht, B.N. & Jakubzick, C.V. Biology of lung macrophages in health and disease. *Immunity* **55**, 1564-1580 (2022).
11. Schyns, J. *et al.* Non-classical tissue monocytes and two functionally distinct populations of interstitial macrophages populate the mouse lung. *Nat Commun* **10**, 3964 (2019).
12. Chakarov, S. *et al.* Two distinct interstitial macrophage populations coexist across tissues in specific subtissular niches. *Science* **363** (2019).
13. Liegeois, M., Legrand, C., Desmet, C.J., Marichal, T. & Bureau, F. The interstitial macrophage: A long-neglected piece in the puzzle of lung immunity. *Cell Immunol* **330**, 91-96 (2018).
14. Schyns, J., Bureau, F. & Marichal, T. Lung Interstitial Macrophages: Past, Present, and Future. *J Immunol Res* **2018**, 5160794 (2018).

15. Skold, M. & Behar, S.M. Tuberculosis triggers a tissue-dependent program of differentiation and acquisition of effector functions by circulating monocytes. *J Immunol* **181**, 6349-6360 (2008).
16. Zheng, W. *et al.* Mycobacterium tuberculosis resides in lysosome-poor monocyte-derived lung cells during chronic infection. *bioRxiv* (2023).
17. Duan, M. *et al.* CD11b immunophenotyping identifies inflammatory profiles in the mouse and human lungs. *Mucosal Immunol* **9**, 550-563 (2016).
18. Caruso, A.M. *et al.* Mice deficient in CD4 T cells have only transiently diminished levels of IFN-gamma, yet succumb to tuberculosis. *J. Immunol.* **162**, 5407-5416 (1999).
19. Scanga, C.A. *et al.* Depletion of CD4(+) T cells causes reactivation of murine persistent tuberculosis despite continued expression of interferon gamma and nitric oxide synthase 2. *The Journal of experimental medicine* **192**, 347-358 (2000).
20. Mogue, T., Goodrich, M.E., Ryan, L., LaCourse, R. & North, R.J. The relative importance of T cell subsets in immunity and immunopathology of airborne Mycobacterium tuberculosis infection in mice. *The Journal of experimental medicine* **193**, 271-280 (2001).
21. Farlik, M. *et al.* Nonconventional initiation complex assembly by STAT and NF-kappaB transcription factors regulates nitric oxide synthase expression. *Immunity* **33**, 25-34 (2010).
22. Utaisincharoen, P. *et al.* Induction of iNOS expression and antimicrobial activity by interferon (IFN)-beta is distinct from IFN-gamma in Burkholderia pseudomallei-infected mouse macrophages. *Clin Exp Immunol* **136**, 277-283 (2004).
23. Kamijo, R. *et al.* Generation of nitric oxide and clearance of interferon-gamma after BCG infection are impaired in mice that lack the interferon-gamma receptor. *J Inflamm* **46**, 23-31 (1995).
24. Kamijo, R. *et al.* Generation of nitric oxide and induction of major histocompatibility complex class II antigen in macrophages from mice lacking the interferon gamma receptor. *Proc Natl Acad Sci U S A* **90**, 6626-6630 (1993).
25. Nandi, B. & Behar, S.M. Regulation of neutrophils by interferon-gamma limits lung inflammation during tuberculosis infection. *J Exp Med* **208**, 2251-2262 (2011).
26. Lovewell, R.R., Baer, C.E., Mishra, B.B., Smith, C.M. & Sasseti, C.M. Granulocytes act as a niche for Mycobacterium tuberculosis growth. *Mucosal Immunol* (2020).
27. Lee, J. *et al.* CD11c^{hi} monocyte-derived macrophages are a major cellular compartment infected by Mycobacterium tuberculosis. *PLoS Pathog* **16**, e1008621 (2020).
28. Mott, D. *et al.* High Bacillary Burden and the ESX-1 Type VII Secretion System Promote MHC Class I Presentation by Mycobacterium tuberculosis-Infected Macrophages to CD8 T Cells. *J Immunol* **210**, 1531-1542 (2023).

29. Yao, Y. *et al.* Induction of Autonomous Memory Alveolar Macrophages Requires T Cell Help and Is Critical to Trained Immunity. *Cell* **175**, 1634-1650 e1617 (2018).
30. Hoft, S.G. *et al.* The Rate of CD4 T Cell Entry into the Lungs during Mycobacterium tuberculosis Infection Is Determined by Partial and Opposing Effects of Multiple Chemokine Receptors. *Infect Immun* **87** (2019).